# ReplaceAnything3D: Text-Guided Object Replacement in 3D Scenes with Compositional Scene Representations

**Edward Bartrum**[*]
University College London
London, England
edward.bartrum.18@ucl.ac.uk

**Thu Nguyen-Phuoc**
Meta Reality Labs
London, England
thunp@meta.com

**Chris Xie**
Meta Reality Labs
Redmond, Washington
chrisdxie@meta.com

**Zhengqin Li**
Meta Reality Labs
Redmond, Washington
zhl@meta.com

**Numair Khan**
Meta Reality Labs
Redmond, Washington
numairkhan@meta.com

**Armen Avetisyan**
Meta Reality Labs
London, England
aavetisyan@meta.com

**Douglas Lanman**
Meta Reality Labs
Redmond, Washington
douglas.lanman@meta.com

**Lei Xiao**
Meta Reality Labs
Redmond, Washington
lei.xiao@meta.com

## Abstract

We introduce ReplaceAnything3D model (RAM3D), a novel method for 3D object replacement in 3D scenes based on users' text description. Given multi-view images of a scene, a text prompt describing the object to replace, and another describing the new object, our Erase-and-Replace approach can effectively swap objects in 3D scenes with newly generated content while maintaining 3D consistency across multiple viewpoints. We demonstrate the versatility of RAM3D by applying it to various realistic 3D scene types, showcasing results of modified objects that blend in seamlessly with the scene without impacting its overall integrity.

## 1 Introduction

The explosion of new media platforms and display devices has sparked a surge in demand for high-quality 3D content, and thus an increasing need for efficient tools for generating and editing them. While there has been significant progress in 3D reconstruction and generation, 3D scene editing remain a less-studied area. In this work, we focus on the task of replacing or adding new 3D objects to an existing 3D scene using only input language prompts from a user. Compared to other 3D scene editing methods such as relighting or stylization, this task involves intricate local edits to seamlessly integrate new objects into the scene without disrupting its overall coherence. This goes beyond just generating realistic visuals and demands a nuanced understanding of both the global scene context and the interaction between the newly added object and the rest of the scene.

Naively using text-to-3D methods to generate 3D objects and manually adding them to a scene can be a tedious process. More importantly, it completely ignores the interaction between objects'

---

[*]Work done during an internship at Meta Reality Labs Research
Project page: https://replaceanything3d.github.io

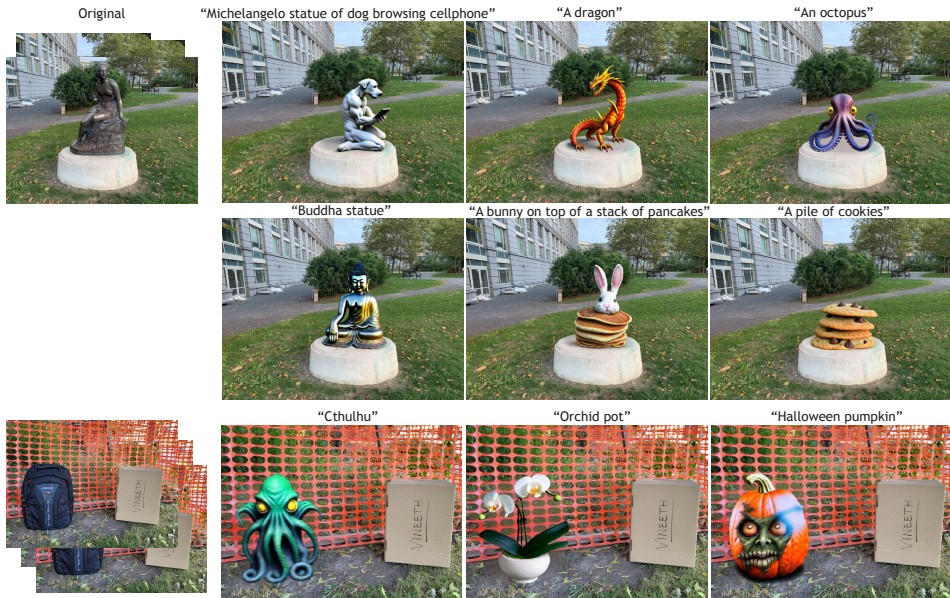

Figure 1: Our method enables prompt-driven object replacement for a variety of realistic 3D scenes.

appearance and the rest of the scene, such as lighting and shadows. Here, we instead formulate the task of object replacement as a 3D scene inpainting problem. Specifically, our goal is to seamlessly fill in the region occupied by the old object with a new object, making it indistinguishable from the rest of the 3D scene. To avoid manual object placement and blending, we adopt a powerful text-guided image inpainting model, enabling 3D object replacement based solely on input text prompts.

In this work, we present the ReplaceAnything3D model (RAM3D), a text-guided method for object replacement in 3D scenes using an Erase-and-Replace strategy. RAM3D takes multiview images of a static scene as input, along with text prompts specifying which object to erase and what should replace it. Our approach comprises four key steps: 1) We use a text-to-mask model with a text prompt to detect and segment the object to be erased from the input images. 2) We use a text-guided 3D inpainting technique to fill in the background region obscured by the removed object in a multi-view consistent manner. 3) Next, we use a similar text-guided 3D inpainting technique to generate a new 3D object corresponding to the input text description, which is seamlessly composited onto the background in all views. We thus obtain multiview consistent images of the edited 3D scene. 4) Finally, these updated dataset images are used to reconstruct the modified 3D scene, enabling novel view synthesis. Instead of relying on generic image editing methods such as Instruct-Pix2Pix [1], we adopt HiFA [2], a state-of-the-art distillation approach, to distill a pretrained text-to-image-inpainting model into a 3D scene representation. This allows us to freely add or entirely remove detailed 3D objects from scenes, a significant challenge for methods like Instruct-Pix2Pix [1] and its derivatives such as Instruct-NeRF2NeRF [3] due to their reliance on a limited dataset of images and editing instructions. By integrating a text-guided image inpainting model with a compositional scene structure that distinguishes the object of interest from the rest of the scene, ReplaceAnything3D can seamlessly generate edited 3D scenes with new objects harmoniously integrated into their environment. In summary, our contributions are:

- We introduce a method for text-guided object replacement in 3D scenes that removes the need for manual 3D modelling.

- We propose a multi-stage approach that supports object replacement or even object addition in 3D scenes in high-fidelity.

- We present 3D consistent results on multiple scene types (human avatar, forward-facing and $360°$ scenes), and challenging edit prompts requiring detailed texture synthesis.

## 2 Related work

**Diffusion model for text-guided image editing**   Diffusion models trained on extensive text-image datasets have demonstrated remarkable results, showcasing their ability to capture intricate semantics from text prompts [4, 5, 6]. As a result, these models provide strong priors for various text-guided image editing tasks [7, 8, 9, 1, 10]. In particular, methods for text-guided image inpainting [11, 12] enable local image editing by replacing masked regions with new content that seamlessly blends with the rest of the image, allowing for object removal, replacement, and addition. These methods are direct 2D counterparts to our approach for 3D scenes, where each view can be treated as an image inpainting task. However, 3D scenes present additional challenges, such as the requirement for multi-view consistency and memory constraints due to the underlying 3D representations. In this work, RAM3D addresses these challenges by combining a pre-trained image inpainting model with compositional 3D scene representations.

**Text-to-3D synthesis**   With the remarkable success of text-to-image diffusion models, text-to-3D synthesis has garnered increasing attention. Most work in this area focuses on distilling pre-trained text-to-image models into 3D models, starting with the seminal works Dreamfusion [13] and Score Jacobian Chaining (SJC) [14]. Subsequent research has explored various methods to enhance the quality of synthesized objects [15, 2, 16, 17] and disentangle geometry and appearance [18]. Instead of relying solely on pre-trained text-to-image models, recent work has utilized large-scale 3D datasets such as Objaverse [19] to improve the quality of 3D synthesis from text or single images [20, 21].

Here, we move beyond text-to-3D synthesis by incorporating both text prompts and the surrounding scene information as inputs. This approach introduces additional complexities, such as ensuring the appearance of the 3D object harmoniously blends with the rest of the scene and accurately modeling object-object interactions like occlusion and shadows. Combining HiFA [2], a text-to-3D distillation approach, with a text-to-image-inpainting model, RAM3D aims to create more realistic and coherent 3D scenes that seamlessly integrate the synthesized 3D objects (Figure 1).

**3D Editing**   Many existing 3D editing methods focus on editing an individual object's appearance or geometry [22, 23, 24, 25, 26]. For scene-level editing, recent works primarily address object removal tasks for forward-facing NeRF scenes [27, 28, 29]. Instruct-NeRF2NeRF [3] and similar followup works [30, 31, 32] offer a comprehensive approach to both appearance editing and object addition, leveraging InstructPix2Pix [1] to update the scene dataset. However, as they modify the entire scene, they struggle to synthesise objects with complex geometrical texture patterns, and completely fail to remove objects from scenes. Blended-Nerf [26] and DreamEditor [33] allow localized object editing but do not support object removal. One closely related work is [28], which can remove and replace objects using one single image reference from the user. However, since this method relies only on a single inpainted image, it cannot handle regions with large occlusions across different views, and thus is only applied on forward-facing scenes. Another closely related work is RePaint-NeRF [34], which similarly allows text-guided scene editing on a masked region, but uses SDS loss to update a pretrained NeRF towards the text content. In contrast, RAM3D adopts an Erase-and-Replace approach for localized scene editing, instead of modifying the existing geometry or appearance of the scene's content, leading to superior qualitative results when compared to RePaint-NeRF.

Unlike NeRF editing methods, the recently proposed Gaussian Editor [35], uses 3D Gaussian Splats [36] as its underlying representation. Exploiting the explicit nature of this representation, this method enables localised edits by selectively updating targeted Gaussians, using guidance from InstructPix2Pix [1]. Like our work, it also supports adding new objects to scenes. Similarly to [28], a single scene image is inpainted in 2D, providing a reference image of the new object. A pre-existing image-to-3D object model [37] then generates a 3D object from the segmented reference image, in isolation from the original scene. The coarse object mesh is transformed back into a 3D Gaussian representation and integrated into the scene through a laborious modelling process where the object is is manually placed into the scene. More importantly, the new object is not guaranteed to integrate seamlessly into the surrounding scene, which results in visible artifacts and quality issues when the method is applied to $360°$ scenes. In contrast, RAM3D performs distillation using the inpainting network as a diffusion prior, which leads to harmonious blending between the new object and its surroundings, even when applied to $360°$ scenes with challenging edit prompts.

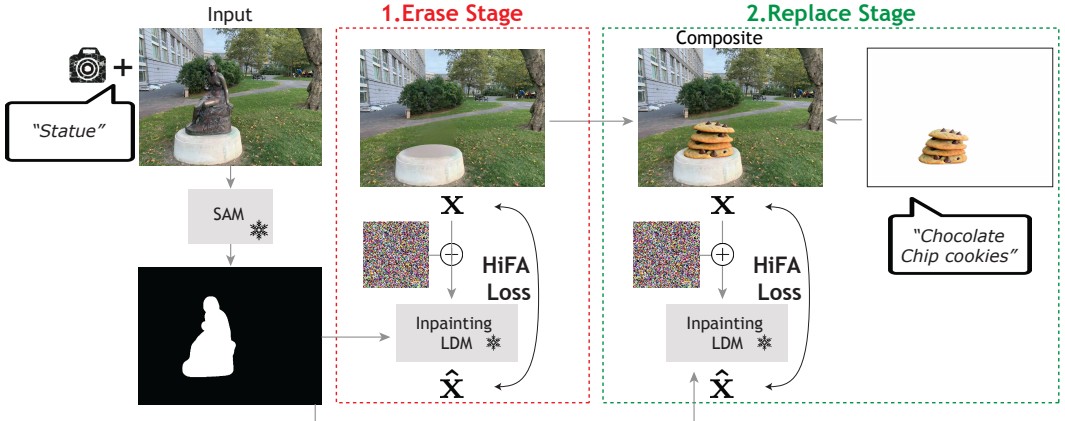

Figure 2: An overview of RAM3D **Erase** and **Replace** stages.

# 3 Method

## 3.1 Preliminary

**Distilling diffusion models** Dreamfusion [13] proposes a technique called Score Distillation Sampling (SDS) to compute gradients from a 2D pre-trained text-to-image diffusion model, to optimize the parameters of 3D neural radiance fields (NeRF). Recently, HiFA [2] significantly improves the quality of text-to-3D object generation by introducing an alternative loss formulation to SDS that can be computed explicitly for a Latent Diffusion Model (LDM). Let $\theta_{\text{scene}}$ be the parameters of a implicit 3D scene, $y$ is a text prompt, $\epsilon_\phi(\mathbf{z_t}, t, y)$ be the pre-trained LDM model with encoder $E$ and decoder $D$, $\theta_{\text{scene}}$ can be optimized using:

$$\mathcal{L}_{\text{HiFA}}(\phi, \mathbf{z}, \mathbf{x}) = \mathbb{E}_{t,\epsilon} w(t) \left[ \|\mathbf{z} - \hat{\mathbf{z}}\|^2 + \lambda_{\text{RGB}} \|\mathbf{x} - \hat{\mathbf{x}}\|^2 \right] \tag{1}$$

where $\mathbf{z} = E(\mathbf{x})$ is the latent vector by encoding a rendered image $\mathbf{x}$ of $\theta_{\text{scene}}$ from a camera viewpoint from the training dataset, $\hat{\mathbf{z}}$ is the estimate of latent vector $\mathbf{z}$ by the denoiser $\epsilon_\phi$, and $\hat{\mathbf{x}} = D(\hat{\mathbf{z}})$ is a recovered image obtained through the decoder $D$ of the LDM. Note that for brevity, we incorporate coefficients related to timesteps $t$ to $w(t)$.

Here we deviate from the text-to-3D synthesis task where the generated object is solely conditioned on a text prompt. Instead, we consider a collection of scene views as additional inputs for the synthesized object. To achieve this, we utilize the HiFA distillation loss function and timestep-annealing strategy, in conjunction with an open-source text-to-image *inpainting* LDM. This LDM $\epsilon_\psi(\mathbf{z_t}, t, y, \mathbf{m})$ requires not only a text prompt $y$, but also a binary mask $\mathbf{m}$ indicating the area to be filled in.

## 3.2 Overview

The input to RAM3D consists of a collection of $n$ images $I_i$, corresponding camera viewpoints $\mathbf{v}_i$ and a text prompt $y_{\text{erase}}$ describing the object the user wishes to replace. Using this text prompt we can obtain masks $\mathbf{m}_i$ corresponding to every image and camera viewpoint using a pretrained text-to-mask model LangSAM [38]. Note that these masks are not necessarily multi-view consistent. We additionally provide a text prompt $y_{\text{replace}}$ describing a new object to replace the old object. Our goal is to modify the masked object in every image in the dataset to match the text prompt $y_{\text{replace}}$, in a 3D-consistent manner. We can then reconstruct the edited scene using any choice of 3D representations such as NeRF [39] or Gaussian Splats [36] to obtain renderings of the edited 3D scene from novel viewpoints.

Figure 2 illustrates the overall pipeline of our **Erase and Replace** framework. Instead of modifying existing objects' geometry and appearance to match the target text descriptions like other methods [3, 33], we adopt an Erase-and-Replace approach. Firstly, for the **Erase** stage, we remove the masked objects completely and inpaint the occluded region in the background, using a neural field $\theta_{\text{bg}}$. Secondly, for the **Replace** stage, we generate new objects using a neural field $\theta_{\text{fg}}$, compositing them so that they blend in with the inpainted background scene. Finally, we create a new training set

using the edited images and camera poses from the original scene, and reconstruct the modified 3D scene using any choice of 3D representations for novel view synthesis.

To enable text-guided object replacement in 3D scenes, we distill an open-source text-to-image inpainting LDM using the HiFA loss function (Equation 1) and timestep annealing strategy [2] . Note that LDM distillation using 3D Gaussian Splats is still challenging to optimise, leading to blurry results and thus requiring further refinement process in recent text-to-3D-object works [40, 41]. We therefore opt to use a NeRF-based representation instead, for RAM3D's **Erase** and **Replace** stages (Sections 3.3, 3.4). To circumvent the memory constraints and slow training speed inherent to NeRF's implicit representations, we propose a Bubble-NeRF representation (see Figure 3, Left side) which only models the localised part of the scene that is affected by the editing operation, instead of the whole scene.

### 3.3 Erase stage

In the Erase stage, we aim to remove the object described by $y_{\text{erase}}$ from the scene and inpaint the occluded background region in a multi-view consistent manner. To do so, we optimise RAM3D parameters $\theta_{\text{bg}}$ which implicitly represent the inpainted background scene. Note that the Erase stage only needs to be performed once to remove the desired object, after which the Replace stage (Section 3.4) can be used to generate different objects or even add new objects to the scene, as demonstrated in the Results section. As a pre-processing step, we use LangSAM [38] with text prompt $y_{\text{erase}}$ to obtain a mask $\mathbf{m}_i$ for each image in the dataset. We then dilate each $\mathbf{m}_i$ to obtain *halo* regions $\mathbf{h}_i$ around the original input mask (see Figure 3, Left side).

At each training step, we sample image $I_i$, camera $\mathbf{v}_i$, mask $\mathbf{m}_i$, and halo region $\mathbf{h}_i$ for a random $i \in \{1..n\}$, providing them as inputs to RAM3D to compute training losses (Figure 2, Left side) (we henceforth drop the subscript i for clarity). RAM3D volume renders the implicit 3D representation $\theta_{\text{bg}}$ over rays emitted from camera viewpoint $\mathbf{v}$ which pass through the visible pixels in $\mathbf{m}$ and $\mathbf{h}$ (the Bubble-NeRF region). The RGB values of the remaining pixels on the exterior of the Bubble-NeRF are sampled from $I$ (see Figure 3, Left side). These rendered and sampled pixel rgb-values are arranged into a 2D array, and form RAM3D's inpainting result for the given view, $\mathbf{x}^{\text{bg}}$. Following the HiFA distillation objective (see Section 3.1), we use the frozen LDM's $E$ to encode $\mathbf{x}^{\text{bg}}$ to obtain $\mathbf{z}^{\text{bg}}$, add noise, denoise with $\epsilon_{\psi}$ to obtain $\hat{\mathbf{z}}^{\text{bg}}$, and decode with $D$ to obtain $\hat{\mathbf{x}}^{\text{bg}}$. We condition $\epsilon_{\psi}$ with $I$, $\mathbf{m}$ and the empty prompt, since we do not aim to inpaint new content at this stage.

We now use these inputs to compute $\mathcal{L}_{\text{HiFA}}$ (see Equation 1). We next compute $\mathcal{L}_{\text{recon}}$ and $\mathcal{L}_{\text{vgg}}$ on $\mathbf{h}$ (see Figure 3), guiding the distilled $\theta_{\text{bg}}$ towards an accurate reconstruction of the background.

$$\mathcal{L}_{\text{recon}} = \text{MSE}(\mathbf{x}^{\text{bg}} \odot \mathbf{h}, I \odot \mathbf{h}) \tag{2}$$

$$\mathcal{L}_{\text{vgg}} = \text{MSE}(vgg_{16}(\mathbf{x}^{\text{bg}} \odot \mathbf{h}), vgg_{16}(I \odot \mathbf{h})) \tag{3}$$

This step is critical to ensuring that RAM3D inpaints the background correctly (see Figure 7). Following [42], we compute depth regularisation $\mathcal{L}_{\text{depth}}$, leveraging the geometric prior from a pretrained depth estimator [43]. In summary, the total loss for the Erase stage is:

$$\mathcal{L}_{\text{Erase}} = \mathcal{L}_{\text{HiFA}} + \lambda_{\text{recon}}\mathcal{L}_{\text{recon}} + \lambda_{\text{vgg}}\mathcal{L}_{\text{vgg}} + \lambda_{\text{depth}}\mathcal{L}_{\text{depth}} \tag{4}$$

### 3.4 Replace stage

In the Replace stage, we aim to add a new object described by $y_{\text{replace}}$ into the inpainted scene. To do so, we optimise the foreground neural field $\theta_{\text{fg}}$ to render $\mathbf{x}^{\text{fg}}$, which is then composited with $\mathbf{x}^{\text{bg}}$ to form $\mathbf{x}$. Unlike $\theta_{\text{bg}}$ in the Erase stage, $\theta_{\text{fg}}$ does not seek to reconstruct the background scene, but instead only the LDM-inpainted object which is located on the interior of $\mathbf{m}$. Therefore in the Replace stage, RAM3D does not consider the halo rays which intersect $\mathbf{h}$, but only those intersecting $\mathbf{m}$ (Figure 3, Right side). These rendered pixels are arranged in the masked region into a 2D array to give the foreground image $\mathbf{x}^{\text{fg}}$, whilst the unmasked pixels are assigned an RGB value of 0. The accumulated densities are similarly arranged into a foreground alpha map $A$, whilst the unmasked pixels are assigned an alpha value of 0. We now composite the foreground $\mathbf{x}^{\text{fg}}$ with the background $\mathbf{x}^{\text{bg}}$ using alpha blending:

$$\mathbf{x} = A \odot \mathbf{x}^{\text{fg}} + (1 - A) \odot \mathbf{x}^{\text{bg}} \tag{5}$$

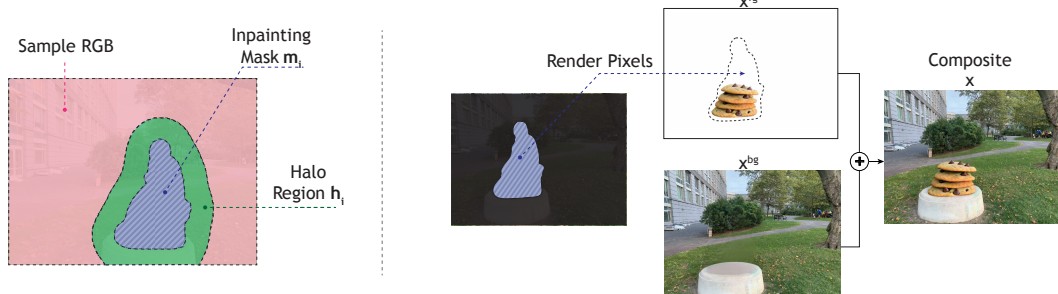

Figure 3: **Left: Erase stage.** The masked region (blue) serves as a conditioning signal for the LDM, indicating the area to be inpainted. The surrounding nearby pixels form the halo region $h$ (green), which is also rendered by RAM3D during the Erase stage. The union of these 2 regions is the Bubble-NeRF region, whilst the remaining pixels are sampled from the input image (red).
**Right: Replace stage.** RAM3D volumetrically renders the masked pixels (shown in blue) to give $\mathbf{x^{fg}}$. The result is composited with $\mathbf{x^{bg}}$ to form the combined image $\mathbf{x}$.

Using the composited result $\mathbf{x}$, we compute $\mathcal{L}_{\text{HiFA}}$ as before, but now condition $\epsilon_\psi$ with the prompt $y_{\text{replace}}$, which specifies the new object for inpainting. As we no longer require the other losses, we set $\lambda_{\text{recon}}, \lambda_{\text{vgg}}, \lambda_{\text{depth}}$ to 0.

Since the Erase stage already provides us with a good background, in this stage, $\theta_{\text{fg}}$ only needs to represent the foreground object. To encourage foreground/background disentanglement, on every $k$-th training step, we substitute $\mathbf{x^{bg}}$ with a constant-value RGB tensor, with randomly sampled RGB intensity. This guides the distillation of $\theta_{\text{fg}}$ to only include density for the new object; a critical augmentation to avoid spurious floaters over the background (see Figure 7, Left side).

## 3.5 Reconstructing the edited scene

Once the inpainted background and objects have been generated inside the Bubble-NeRF region (Figure 3) during the Erase and Replace stages, we composite the Bubble-NeRF renderings onto all original scene images. We finally obtain a full 3D representation of the edited scene by applying an off-the-shelf scene reconstruction method such as NeRF or Gaussian splats [36].

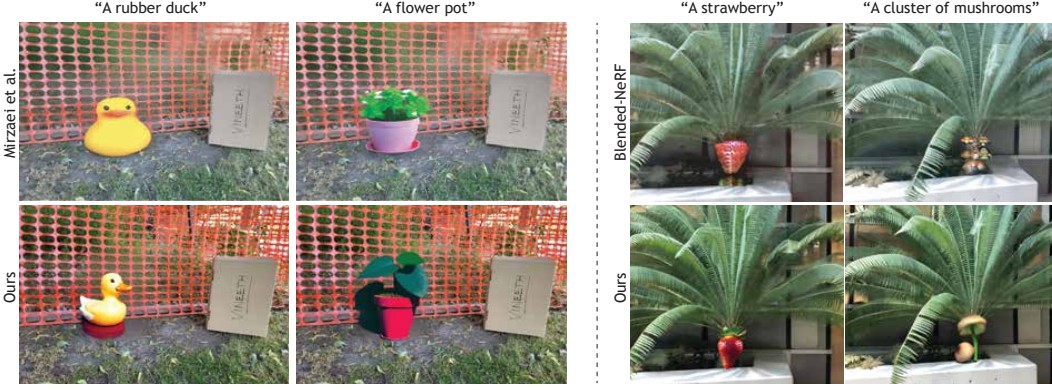

Figure 4: **Left:** Qualitative comparison with Reference-Guided Inpainting [28] (images adapted from the original paper) for object replacement .
**Right:** Qualitative comparison with Blended-NeRF [12] for object replacement. Our method generates results with higher quality and capture more realistic lighting and details.

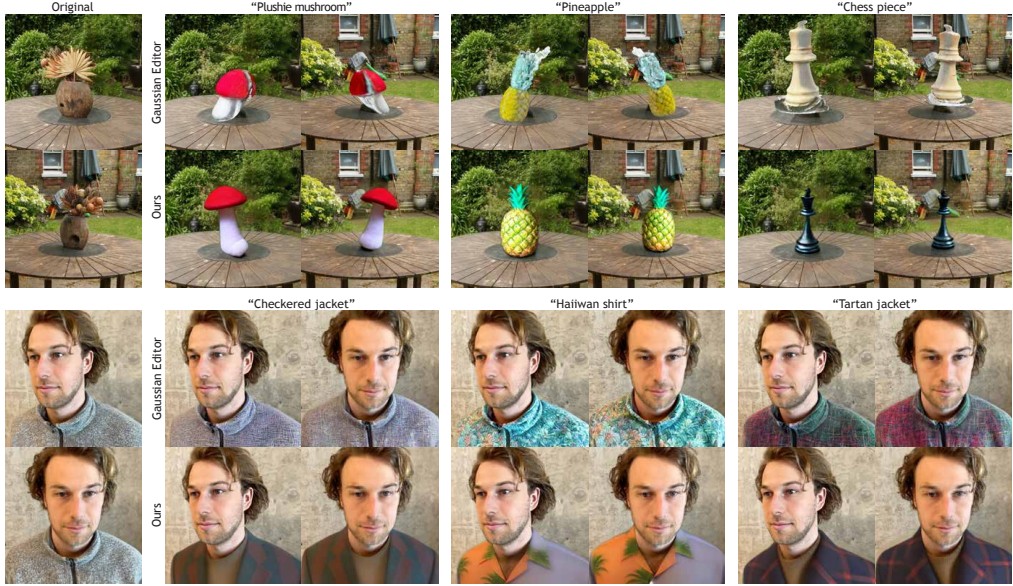

Figure 5: Qualitative comparison with Gaussian Editor [35]. We show results for 3 challenging edit prompts on the GARDEN scene (top 2 rows) and FACE scene (bottom 2 rows). In the GARDEN scene, our method generates more realistic objects which are better integrated with the surrounding scene. In the FACE scene, our method generates more detailed texture patterns and geometry which are better aligned with the edit prompts.

## 4 Results

We conduct experiments on real 3D scenes varying in complexity: forward-facing scenes, 360° scenes and human avatar. For forward-facing scenes, we show results for the STATUE and RED-NET scene from SPIn-NeRF dataset [27], as well as the FERN scene from NeRF [39]. For 360° scene, we show results from the GARDEN scene from Mip-NeRF 360°[44]. For the avatar result, we use the FACE dataset from Instruct-NeRF2NeRF [3]. On each dataset, we train RAM3D with a variety of $y_{\text{replace}}$, generating a diverse set of edited 3D scenes (Figure 1). Please refer to the supplemental video for more qualitative results for object replacement using personalized content.

Note that RAM3D performs localised scene editing for 3D object replacement, conditioned by an edit region mask and replacement object description prompt, as shown in Figure 2. For fair apples-to-apples comparison, we mostly compare with other state-of-the-art localised editing methods (see Figure 4, Figure 5, Table 1). This scope contrasts with the separate but related track of global scene editing methods, which can modify an entire scene based on instruction-style prompts, but do not support object replacement or removal [3, 30, 31, 32]. We nevertheless provide qualitative and quantitative comparison with global-editing methods in the Appendix (Sections F, G).

### 4.1 Qualitative Comparisons

Figure 4 shows qualitative comparison with two methods for NeRF-based 3D object replacement; Blended-NeRF [12] and the method by [28]. In Figure 5 we compare our method with a state-of-the-art 3D Gaussian Splat-based scene editing framework [35], which supports object removal, addition and replacement. In the Appendix (Section F), we additionally compare with DreamEditor [33], Repaint-NeRF [34], generic scene editing method Instruct-NeRF2NeRF [3] and various similar InstructPix2Pix-based [1] followup works [30, 31, 32].

Compared to method by [28] on the left side of Figure 4, RAM3D achieves comparable object replacement results while handling more complex lighting effects such as shadows between the foreground and background objects. Note that the method by [28] only works with forward facing scenes, and thus cannot handle 360° scenes such as the GARDEN scene like our method. In Figure

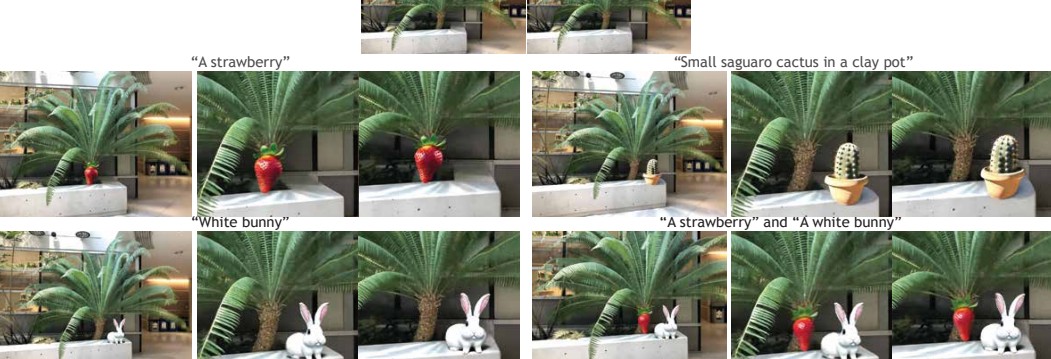

Figure 6: Given user-defined masks, ReplaceAnything3D can add completely new objects that blend in with the rest of the scene. Furthermore, due to its compositional structure, RAM3D can add multiple objects to 3D scenes while maintaining realistic appearance, lighting, and multi-view consistency (bottom right).

4 right side, we note that RAM3D generates more realistic and detailed objects that blend in better with the rest of the scene. Meanwhile, Blended-NeRF only focuses on synthesizing completely new objects without taking the surrounding scenes into consideration. The synthesized object therefore looks saturated and outlandish compared to the rest of the scene. Moreover, due to the memory constraint of CLIP [45] and NeRF, Blended-NeRF only works with image resolutions 2-times smaller than ours ($1008 \times 756$ vs. $504 \times 378$).

Figure 5 shows qualitative comparison with GaussianEditor [35], a state-of-the-art scene-editing framework which supports object deletion, addition, and general editing capabilities. Applying it to the GARDEN scene, we first use the delete functionality to remove the vase from the table, followed by the addition functionality to insert the new object. Note that this method generates the new object in isolation, using a prior image-to-3D method [37], resulting in visible artifacts on the surface of the table where the new object interacts with the surrounding scene. Furthermore, since this method is not guaranteed to place the new object correctly, its position in the scene requires post-hoc manual adjustment - see Appendix (Section H) for further details.

Applying GaussianEditor to the FACE scene, we use the general editing functionality. The explicit Gaussian Splat formulation allows edits to be localised to the relevant Gaussians (corresponding to the man's torso), which is selected using a user interface. The editing process is then guided towards the desired prompt using 2D guidance from InstructPix2Pix [1], and we observe similar limitations to other InstructPix2Pix-based scene-editing methods [3, 32, 31, 30] (see Appendix F). In particular, we observe that GaussianEditor struggles to synthesise detailed texture patterns on the man's torso, unlike RAM3D. Furthermore, we notice that the geometry of the man's clothes appears unchanged in the GaussianEditor results, whereas our method successfully generates geometrical details such as the shirt collar and jacket lapels, matching the edit prompt in each case.

**Adding multiple objects** In addition to replacing objects in the scene, our method can add new objects based on users' input masks. Figure 6 demonstrates that completely new objects with realistic lighting and shadows can be generated and composited to the current 3D scene. Notably, as shown in Figure 6-bottom right, our method can add more than one object to the same scene while maintaining realistic scene appearance and multi-view consistency.

## 4.2 Quantitative Results

3D scene editing is a highly subjective task. Thus, we mainly show qualitative results and comparisons, and refer readers to the supplemental video for additional results. However, we follow [3] and report CLIP Text-Image Direction Similarity, which measures the alignment of the performed object replacement with the input text description. Additionally, we also quantitatively measure temporal

Table 1: We compute a CLIP-based alignment metric, and optical flow-based temporal consistency metric for various datasets and prompts. RAM3D shows the best overall edit prompt alignment and temporal consistency. (Top) GARDEN, (Middle) FACE, (Bottom) FERN.

| Prompts | CLIP Text-Image Direction Similarity ↑ | | Warping error ($\times 10^{-2}$) ↓ | |
| --- | --- | --- | --- | --- |
| | Ours | GaussianEditor | Ours | GaussianEditor |
| Pineapple | **0.2246** | −0.0631 | **1.2600** | 1.4600 |
| Chess | 0.0874 | **0.1857** | **1.2400** | 1.6700 |
| Mushroom | **0.1289** | 0.1030 | **1.2000** | 1.3900 |
| Popcorn | **0.2024** | 0.0400 | **1.3000** | 1.6800 |
| Checker | **0.0446** | 0.0016 | **0.4300** | 0.7600 |
| Hawaiian | **0.2169** | 0.1689 | **0.4900** | 0.8100 |
| Tartan | **0.1015** | 0.0379 | **0.4700** | 0.6700 |
| | Ours | BlendedNerf | Ours | BlendedNerf |
| Mushroom | **0.0928** | 0.0535 | **2.3900** | 2.9500 |
| Strawberry | **0.3165** | 0.2224 | **2.3800** | 3.2300 |

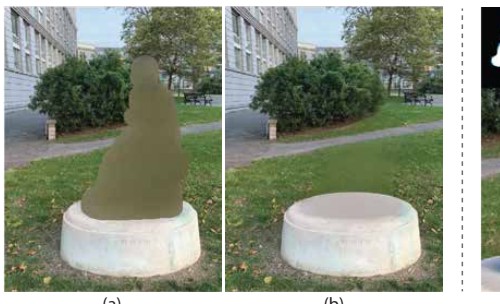 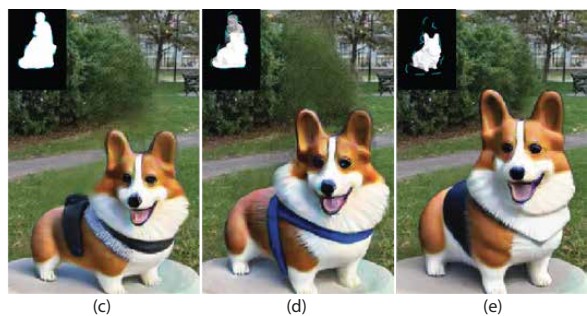

(a)      (b)      (c)      (d)      (e)

Figure 7: **Left:** Results for 2 RAM3D variants trained on the Statue scene for the Erase stage. a) Training without any supervision on the halo region surrounding the inpainting mask. The training objective is ambiguous and the Bubble-NeRF model collapses to a hazy cloud. b) Adding halo losses ($\mathcal{L}_{\text{recon}}$ and $\mathcal{L}_{\text{vgg}}$) for the halo region surrounding the Bubble-NeRF guides the distillation of $\theta_{\text{bg}}$ towards the true background, as observed on rays which pass nearby to the occluding object. RAM3D can now inpaint the background scene accurately.
**Right:** Results for 3 RAM3D variants, on the statue scene for prompt *"A corgi"*. RGB samples are shown with accumulated NeRF density (alpha map) in the top-left corner. The bubble rendering region is shown as a dotted blue line. c) A monolithic scene representation which contains both the foreground and background. d) A compositional scene model but without random background augmentation. e) Our full model.

consistency by calculating the Warping Error, following [46, 47, 48]. Specifically, **(1)** we use RAFT [49] to calculate the optical flow of test videos, with each frame being a rendered novel view of the original scene, **(2)** warp the corresponding frames from the modified scene according to it, and **(3)** measure the warping error. We compare RAM3D quantitatively with 2 state-of-the-art NeRF-based and Gaussian-Splatting-based methods ([35], [50]) for the object-replacement task on three datasets.

In Table 1, we show that RAM3D achieves better overall prompt alignment (highest CLIP Text-Image Direction Similarity) than existing works, and the best temporal consistency (lowest warping error) across all methods and datasets. Interestingly, although Blended-NeRF directly optimizes for CLIP-similarity between the generated objects and target text prompts, it still achieves a lower score than our method. Furthermore, we observe that our model's superior appearance synthesis quality (when compared to GaussianEditor in Figure 5) is also reflected in the quantitative results in Table 1.

### 4.3 Ablation studies

We conduct a series of ablation studies to demonstrate the effectiveness of our method and training strategy. In Figure 7 Right side, we show the benefits of our compositional foreground/background structure and background augmentation training strategy. Specifically, we train a version of RAM3D using a monolithic NeRF to model both the background and the new object (combining $\theta_{bg}$ and $\theta_{fg}$). In other words, this model is trained to edit the scene in one single stage, instead of separate Erase and Replace stages. We observe lower quality background reconstruction in this case, as evident from the blurry hedge behind the corgi's head in Figure 7c.

We also demonstrate the advantage of using random background augmentation in separating the foreground object from the background (see Section 3.4). Without this augmentation, the model is unable to accurately separate the foreground and background alpha maps, resulting in a blurry background and floaters that are particularly noticeable when viewed on video (Figure 7d). In contrast, our full composited model trained with background augmentation successfully separates the foreground and background, producing sharp results for the entire scene (Figure 7e).

In Figure 7 Left side, we show the importance of the Halo region supervision for the Erase stage. Without it, our model lacks important nearby spatial information, and thus cannot successfully generate the background scene.

## 5   Conclusion

We introduce RAM3D, a text-guided 3D object replacement method for 3D scenes, offering a potential editing tool for VR/MR, gaming, and film production. With an Erase-and-Replace approach, RAM3D can effectively replace objects with significantly different contents that blends seamlessly with the original 3D scene. Our method can also add new objects while maintaining realistic appearance and multi-view consistency. We demonstrate the effectiveness of RAM3D in various realistic 3D scenes (including human avatar, forward-facing and $360°$ scenes), and superior synthesis quality compared to current state-of-the-art NeRF and Gaussian Splatting based methods.

For future work, our Bubble-NeRF method could be extended to other representations such as 3D Gaussian splats [36], similar to DreamGaussian [40]. Other interesting future directions include disentangling geometry and appearance to enable more fine-grained control for scene editing, addressing multi-face problems using prompt-debiasing methods [51] or models that are pre-trained on multiview datasets [52, 53], and developing amortized models for faster editing, similar to [54].

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

## A    Appendix

In this appendix, we first provide an Impact statement in Section B. We next provide the results of a User Study in Section C. We provide additional results by our ReplaceAnything3D model (RAM3D) in Section E, as well as additional comparisons with other methods in Sections F and G. In Section D, we show additional ablation results to further validate our method. We give further details on running the GaussianEditor comparisons in Section H. We provide implementation details in Section I. Finally, we discuss current limitations of our method in Section J.

In the video submission, we show additional qualitative results and comparisons, which better showcase the multi-view consistency of the edited scenes by RAM3D.

## B    Impact Statements

Our work contributes to the advancement of 3D generative machine learning, a field with significant ethical considerations, such as the potential for the creation and dissemination of deceptive or manipulated visual content. We acknowledge these concerns and emphasize our commitment to responsible research and development practices. We believe that this technology has the potential to empower professionals in creative industries and individuals seeking to create 3D content responsibly, and we are dedicated to addressing ethical challenges in its deployment.

## C    User Study

We performed a User Study[†] comparing our results with 3 closely related scene works; Gaussian Editor [35], BlendedNeRF [50] and InstructNeRF2NeRF [3]. 15 participants were asked to compare the results from RAM3Dwith these models on a variety of scenes and prompts. Users were asked to choose the result which best matches the input prompt, and which shows the highest visual quality. We report the preference rates for each model in Tables 2 and 3; note that our model's results are preferred overall across both categories.

Table 2: User Study: GARDEN scene and FACE scene

| Criterion | Prompt Matching (3 d.p) ↑ | | | Visual Quality (3 d.p) ↑ | | |
|---|---|---|---|---|---|---|
| Prompt | Ours | IN2N | GaussianEditor | Ours | IN2N | GaussianEditor |
| Pineapple | **0.867** | 0.067 | 0.067 | **0.867** | 0.067 | 0.067 |
| Plushie Mushroom | **0.933** | 0.000 | 0.067 | **0.933** | 0.000 | 0.067 |
| Chess | **1.000** | 0.000 | 0.000 | **1.000** | 0.000 | 0.000 |
| Popcorn | **1.000** | 0.000 | 0.000 | **1.000** | 0.000 | 0.000 |
| Checkered Jacket | **0.800** | 0.000 | 0.200 | **0.530** | 0.067 | 0.400 |
| Hawaiian Shirt | **0.667** | 0.000 | 0.333 | 0.267 | 0.067 | **0.667** |
| Tartan Jacket | **0.733** | 0.067 | 0.200 | **0.533** | 0.000 | 0.467 |

Table 3: User Study: FERN scene

| Criterion | Prompt Matching (3 d.p) ↑ | | Visual Quality (3 d.p) ↑ | |
|---|---|---|---|---|
| Prompt | Ours | BlendedNeRF | Ours | BlendedNeRF |
| Mushrooms | **0.533** | 0.467 | **0.533** | 0.467 |
| Strawberry | **0.800** | 0.200 | **0.800** | 0.200 |

## D    Additional ablation studies

We performed additional ablation studies to validate our model design (considering both Erase and Replace training stages), including testing the importance of HiFA and Depth loss terms, and Halo-region supervision.

---

[†]All activities relating to this User Study took place whilst Edward Bartrum was a visiting student at KAIST; respondents were KAIST researchers.

In Figure 8 we show Replace-stage results using simple SDS loss (with random diffusion timestep sampling), adding a corgi in the statue scene. The results show slightly less detailed texture synthesis as the RGB-space HiFA loss component is removed. Nevertheless, note that our pipeline still synthesises the new object in the correct position, which is orthogonal to synthesis quality. We additionally provide quantitative ablation results in Table 5, which show that using simple SDS loss leads to only a slight drop in CLIP Text-Image Direction Similarity, whilst removing background augmentation or training our model in a single stage also lead to worse performance.

In Figure 9 we show the performance of our model on the Erase-stage task, with Halo supervision and Depth loss components removed, and compare these variants to our full model. We run a state-of-the-art image segmentation model [55] on each model variant's results, to detect the main segmentation mask inside a bounding box around the statue region. As shown in Figure 9, (purple region), the original statue segmentation mask is still detected for our No Halo and No Depth loss model variants. However, for our full model, the statue mask is not detected; SAM instead correctly detects the hedge region behind the statue. This implies that our full method has successfully removed the statue, and realistically filled in the background, including the disoccluded region of the hedge. For a quantitative comparison of these model variants, see Table 4.

|  | No Halo | No Depth loss | Full |
|---|---|---|---|
| CLIP Sim. ↑ | 0.020 | 0.087 | **0.145** |

Table 4: Erase-stage ablation results. We report CLIP Text-Image Direction Similarity scores for all model variants, using the prompt "A white plinth in a park, in front of a path", on the STATUE scene. Note that our full model performs best.

|  | SDS | 1 stage | No BG aug | Full |
|---|---|---|---|---|
| CLIP Sim. ↑ | 0.217 | 0.218 | 0.214 | **0.232** |

Table 5: Replace-stage ablation results. We report CLIP Text-Image Direction Similarity scores for all model variants, using the prompt "A corgi on a white plinth", on the STATUE scene. Note that our full model performs best.

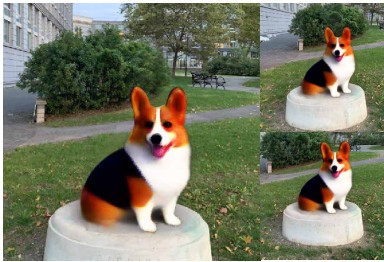

Figure 8: Corgi in STATUE scene, using SDS loss instead of HiFA

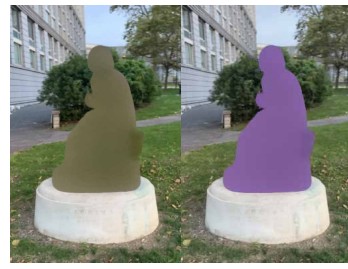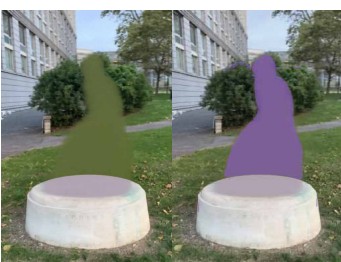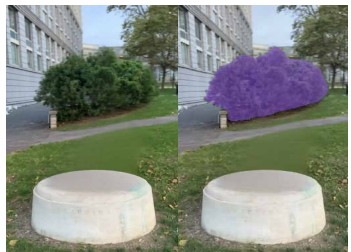

Figure 9: STATUE Erase stage ablation: (SAM segmentation shown in purple). From Left to Right: No Halo supervision, No Depth Loss, Full model - which successfully removed the original statue.

## E  Personalized ReplaceAnything3D

In addition to text prompts, RAM3D enables users to replace or add their own assets to 3D scenes. This is achieved by first fine-tuning a pre-trained inpainting diffusion model with multiple images of a target object using Dreambooth [56]. The resulting fine-tuned model is then integrated into RAM3D to enable object replacement in 3D scenes. As shown in Figure 10, after the fine-tuning stage, RAM3D can effectively replace or add objects to new 3D scenes.

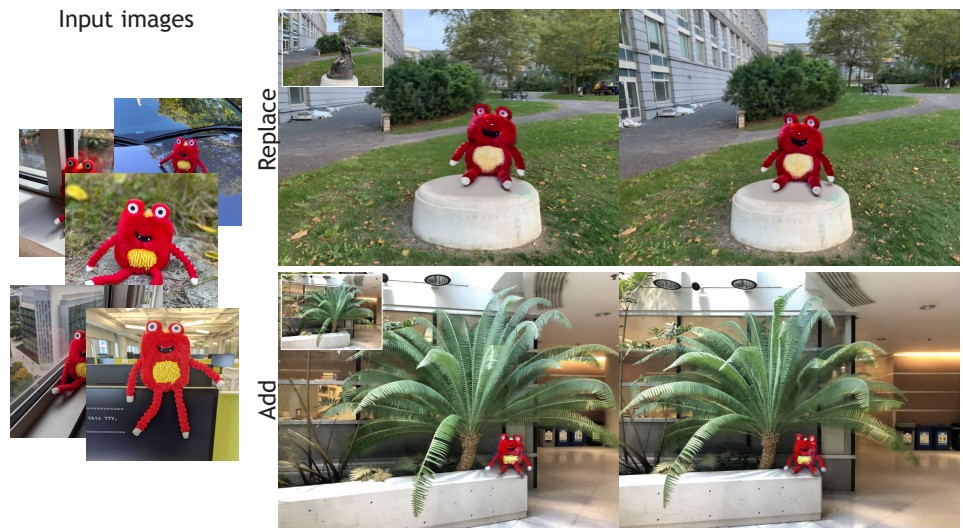

Figure 10: Users can personalize a 3D scene by replacing or adding their own assets using a fine-tuned RAM3D. We achieve this by first fine-tuning an inpainting diffusion model with five images of the target object (left), and then combining it with RAM3D to perform object replacement and addition with custom content.

## F  Additional qualitative comparisons

In Figure 14, we compare our approach with a naive 2D baseline where each image is processed individually. For each image in the training set (first column), we mask out the foreground object (*statue*) and replace it with a new object (*corgi*) using a pre-trained text-to-image inpainting model (Figure 14-second column). We then train a NeRF scene with these modified images. As shown in Figure 14-third column, this results in a corrupted, inconsistent foreground object since each view is very different from each other, in contrast to our multi-view consistent result.

In Figure 11, we compare RAM3D with Instruct-NeRF2NeRF [3], a general scene-editing framework. We note that this method struggles to handle cases where the new object is significantly different from the original one (for example, replace a vase and flowers with a pineapple or a chess piece in Figure 11 second and third column). RAM3D can also generate objects with challenging texture such as the tartan pattern on the jacket, while Instruct-NeRF2NeRF struggles due to its naive reliance on inconsistent dataset updates from Instruct-Pix2Pix [1]. More importantly, Instruct-NeRF2NeRF significantly changes the global structure of the scene even when the edit is supposed to be local (for example, only change the clothing). This is undesirable, especially for human face editing where the rest of the identity should be maintained after the edits. Finally, RAM3D is capable of removing objects from the scene completely, whilst Instruct-NeRF2NeRF cannot (Figure 11 column 1).

In Figure 12 we additionally show FACE dataset results obtained using 3 followup works to Instruct-NeRF-to-NeRF [31, 32, 30]. These methods also rely on InstructPix2Pix [1] to provide a 2D image editing prior, and could not successfully edit the scenes in our experiments, resulting in poor image quality as shown in Figure 12. We note that all of these methods are limited by the paired training dataset of text editing instructions and images before/after the edit that was used to train InstructPix2Pix. GaussianEditor, when used in general editing mode, also faces the same limitation as it relies on InstructPix2Pix to update the selected gaussians, as demonstrated in the FACE dataset

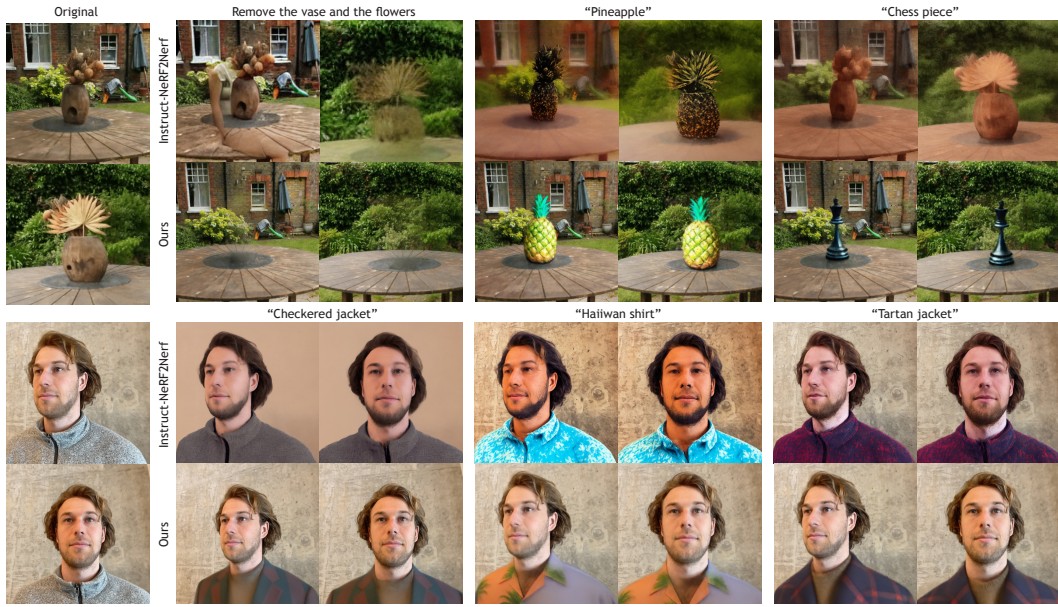

Figure 11: Comparison with Instruct-NeRF2NeRF, a general scene-editing framework [3]. Note that unlike our method, Instruct-NeRF2NeRF modifies the entire scene, cannot synthesise complex texture patterns in the FACE scene and completely fails to generate a pineapple or chess piece object in the 360° GARDEN scene.

Table 6: CLIP-based metrics for GARDEN and FACE datasets, comparing our method with [3]

| Prompts | CLIP Text-Image Direction Similarity ↑ | | CLIP Direction Consistency ↑ | |
|---|---|---|---|---|
| | Ours | InstructNeRF2NeRF | Ours | InstructNeRF2NeRF |
| Pineapple | **0.2041** | 0.0661 | 0.9590 | **0.9660** |
| Chess | **0.1200** | 0.0061 | 0.9457 | **0.9705** |
| Checker | **0.0736** | −0.0400 | 0.9909 | **0.9931** |
| Hawaiian | **0.1805** | 0.1436 | **0.9908** | 0.9884 |
| Tartan | **0.0953** | 0.0714 | **0.9914** | 0.9897 |

results (Figure 5). In contrast to these methods, RAM3D distills a state-of-the-art image-inpainting diffusion model to provide a 2D generative prior, which has been finetuned for inpainting real images, and consequently provides better 2D guidance for the object replacement scene editing task.

In Figure 13 Left side, we present qualitative comparison with RepaintNeRF [34], for object replacement using the NeRF Fortress scene. Note that the Erase-and-Replace approach and Bubble-NeRF formulation adopted by RAM3D can generate higher fidelity novel content, with a more complete shape for the apple. In contrast, note that RePaint-NeRF generates an incomplete shape which appears to be partially cut off by the boundary of the fortress in the input scene. We hypothesise that this artifact is a consequence of the training procedure used by RepaintNeRF, in which the entire input scene is modified directly with diffusion guidance.

In Figure 13 Right side, we demonstrate our model's competitive performance with DreamEditor [10]. It is important to note that DreamEditor has limitations in terms of handling unbounded scenes due to its reliance on object-centric NeuS [57]. Additionally, since DreamEditor relies on mesh representations, it is not clear how this method will perform on editing operations such as object removal, or operations that require significant changes in mesh topologies.

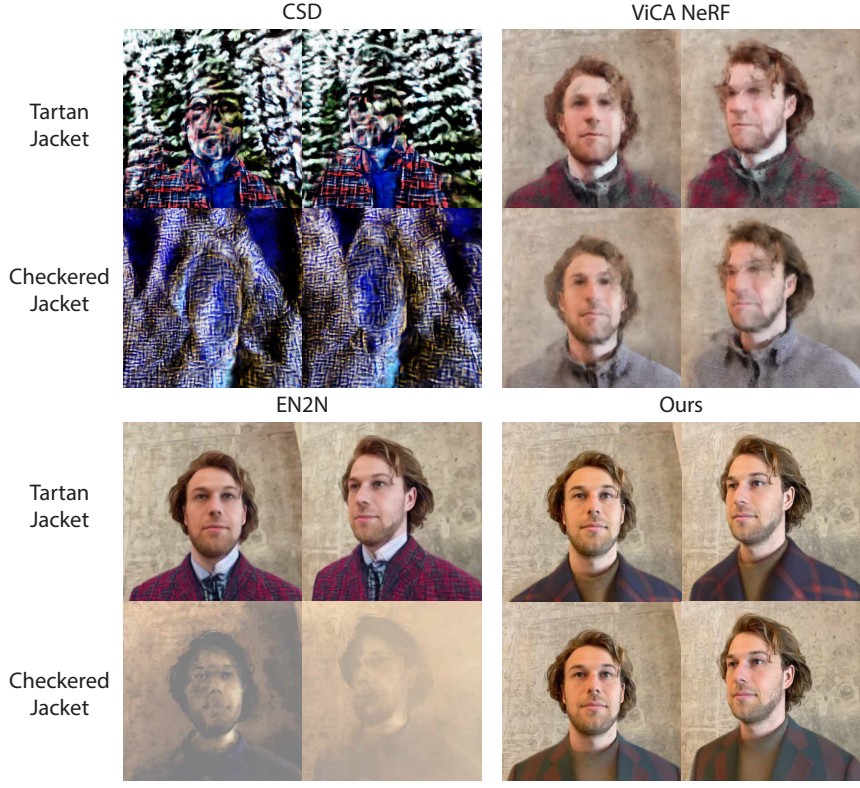

Figure 12: Qualitative comparison with Collaborative Score Distillation (CSD) [32], ViCA-NeRF [31] and EfficientNeRF2NeRF [30]. All approaches apart from our RAM3D were unsuccessful in producing results with intricate texture details for both the checkered and tartan jackets. Results were obtained using official publicly available implementations. Note that in the case of CSD, there is no officially released 3D editing implementation at the time of writing. We therefore followed the official instructions to incorporate CSD image edits into the Instruct-NeRF2NeRF framework.

## G    Additional quantitative comparisons

In Table 6, we provide further quantitative results, comparing our method to Instruct-NeRF2NeRF [3]. We follow [3], reporting the same metrics for both CLIP Text-Image Direction Similarity and CLIP Direction consistency (instead of warping error using optical flow in Section 4.2). However, we note that Instruct-NeRF2NeRF sometime scores higher for CLIP Direction consistency than our method on edit prompts where it completely fails (see Figure 11). For example, for the edit "checkered jacket" in the FACE dataset, Instruct-NeRF2NeRF not only fails to create the checker pattern but also removes high-frequency details in the background, resulting in a solid color background. We hypothesize that this boosts the CLIP-based temporal consistency score even when the edit is unsuccessful. Therefore, we refer readers to the comparisons in the supplemental video for more details.

Note that there are minor variations in the CLIP Text-Image Direction Similarity scores that we report for RAM3D between Tables 1 and 6. This discrepancy is due to a requirement to evaluate GaussianEditor [35] and InstructNeRF2NeRF [3] results on 2 different rendering paths, as these 2 models are trained with different camera conventions (OPENCV vs PINHOLE cameras).

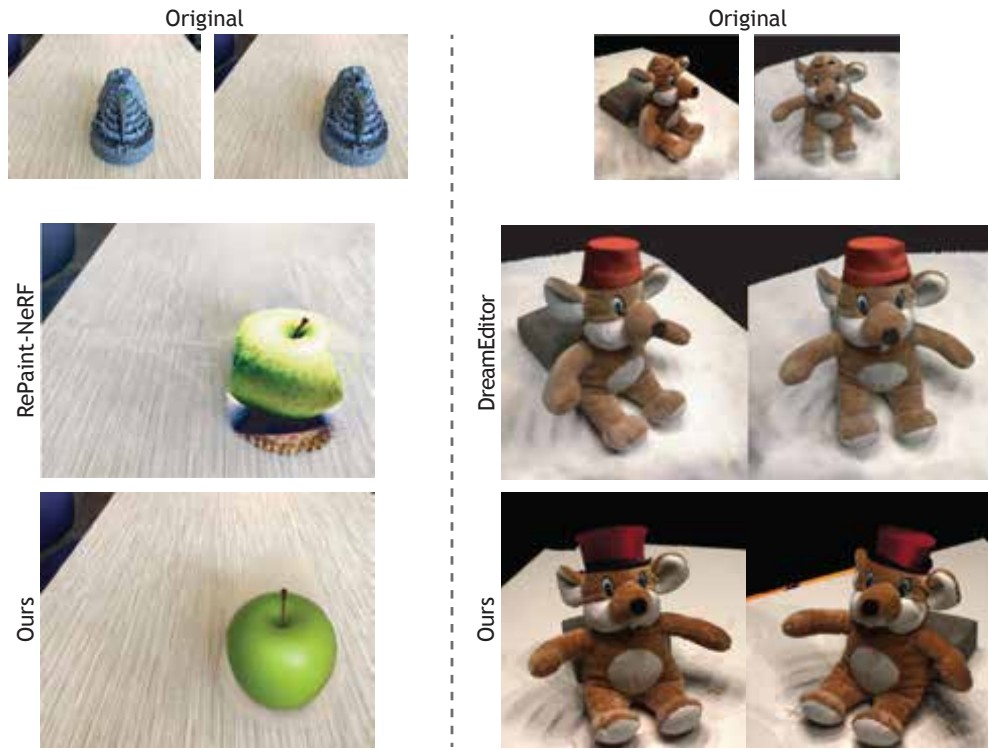

Figure 13: **Left:** Qualitative comparison with RepaintNeRF [34] for object replacement. Figure adapted from the original RepaintNeRF paper.
**Right:** Qualitative comparison with DreamEditor [10] for object addition ("Add a red top hat"). Figure adapted from the original DreamEditor paper.

## H    Gaussian Editor Framework Limitations

We use the official, publicly available GaussianEditor code implementation to obtain results on the GARDEN and FACE scene shown in Figure 5. When obtaining results for the GARDEN scene (top row) we first use the Delete functionality to remove the vase from the table. However, we found that we were unable to replicate the quality of the object removal results reported in the GaussianEditor paper [35]. In particular, we found that the mask-dilation and hole-fixing refinement phase of the Delete method resulted in a cloudy semi-transparent artifact in the area occupied by the removed gaussians (we show the before/after results for this refinement stage in Figure 15). Therefore, in order to compare with the strongest possible baseline, we opted to use the Deletion result from before this refinement step (shown on the left side of Figure 15).

We subsequently use the GaussianEditor Add functionality to incorporate the new object into the scene. We found that the new object was initially placed far from the table by GaussianEditor, which provides a user interface for repositioning the object along the z-values of a camera ray passing through the centre of the reference image. We show the before/after results of this manual depth adjustment in Figure 16, when adding the pineapple object onto the table.

## I    Implementation details

### I.1    NeRF architecture

We use an Instant-NGP [58] based implicit function for the RAM3D NeRF architecture, which includes a memory- and speed-efficient Multiresolution Hash Encoding layer, together with a 3-layer

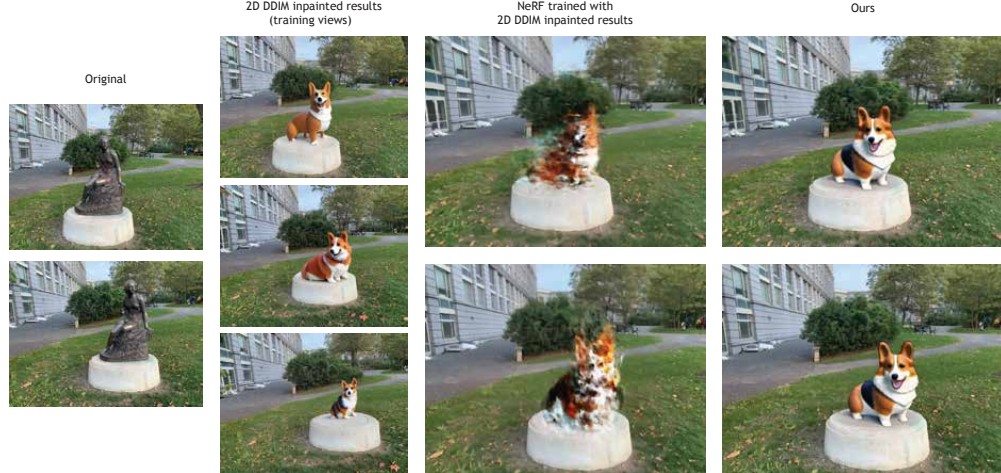

Figure 14: Qualitative comparisons between our method RAM3D (last column) with a naive 2D baseline method, which produces view-inconsistent results (third column). This is because each input image is processed independently and thus vary widely from each other (second column).

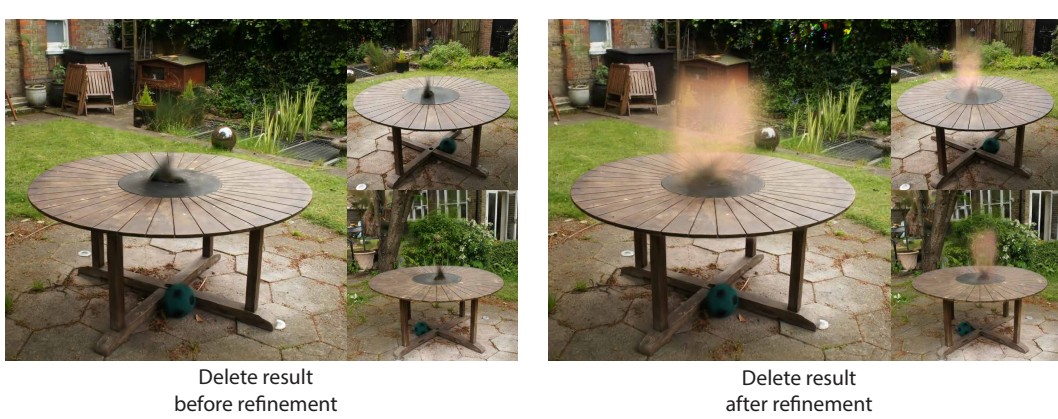

Figure 15: Results obtained using GaussianEditor to remove the vase object from the GARDEN scene. Note that GaussianEditor's proposed mask dilation and hole-fixing refinement stage causes a cloudy artifact to appear above the table, where the vase was originally placed. We therefore use the result on the left, prior to refinement, when adding new objects to replace the vase.

MLP, hidden dimension 64, which maps ray-sample position to RGB and density. We do not use view-direction as a feature. NeRF rendering code is adapted from the nerf-pytorch repo [59].

## I.2 Monolithic vs Erase+Replace RAM3D

We use a 2-stage Erase-and-Replace training schedule for the STATUE, RED-NET and GARDEN scenes. For the FERN scene, we use user-drawn object masks which cover a region of empty space in the scene, therefore object removal is redundant. In this case, we perform object addition by providing the input scene-images as background compositing images to RAM3D. For the FACE scene, the torso region in which we aim to generate new content overlaps entirely with the mask region - we therefore train a monolithic RAM3D to inpaint new content in the mask in a single training stage.

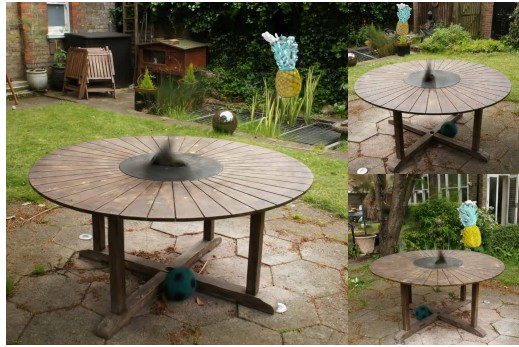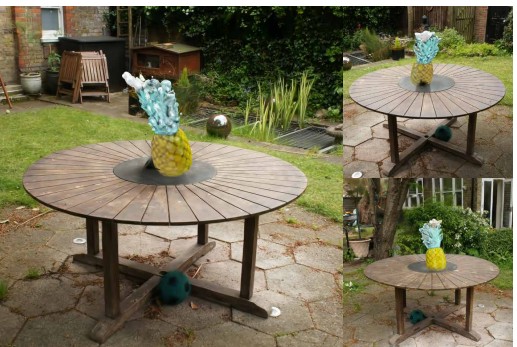

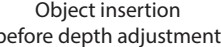

| Object insertion before depth adjustment | Object insertion after depth adjustment |

Figure 16: We observe that the Object Adding functionality of GaussianEditor does not place new objects into the scene in the correct position. The images on the left show the initial pineapple object placement (far from the table), output by GaussianEditor. The images on the right show the results after manually refining the position of the pineapple, by adjusting its depth along a ray passing through the centre of the reference image.

## I.3 Input Masks

We obtain inpainting masks for object removal by passing dataset images to an off-the-shelf text-to-mask model [38], which we prompt with 1-word descriptions of the foreground objects to remove. The prompts used are: STATUE scene: "statue", GARDEN scene: "Centrepiece", FACE scene: "Clothes", RED-NET scene: "Bag". We dilate the predicted masks to make sure they fully cover the object.

For the Erase stage, we compute nearby pixels to the exterior of the inpainting mask, and use them as the Halo region (see Figure 3). We apply reconstruction supervision on the Halo region as detailed in I.5. For the object-addition experiments in the FERN scene, we create user-annotated masks in a consistent position across the dataset images, covering an unoccupied area of the scene.

In Figure 17, we show an example of the input masks, which can be obtained through either using LangSAM and a text prompt to describe the object to be replaced or by manual user placement to add more objects to the 3D scene.

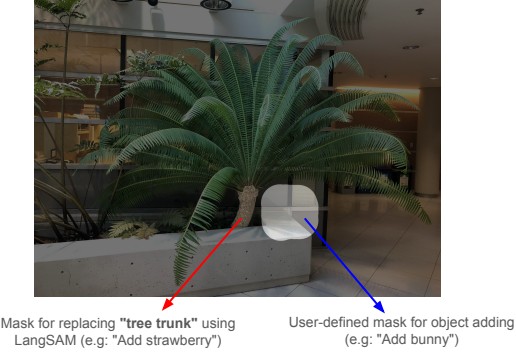

Mask for replacing **"tree trunk"** using LangSAM (e.g: "Add strawberry")     User-defined mask for object adding (e.g: "Add bunny")

Figure 17: Examples of input masks: the red arrow indicates examples of input masks for object replacement, while the blue arrow shows examples of input masks for object addition.

## I.4 Cropping the denoiser inputs

The LDM denoising U-net takes input images of size 512×512. In contrast, RAM3D outputs are of equal resolution to the input scene images, which can be non-square. To ensure size compatibility, we need to crop and resize the RAM3D outputs to 512×512 before passing them to the denoiser (see

Figure 2). For the STATUE, FACE and GARDEN scenes, we resize all images to height 512 and take a centre-crop of 512×512, which always contains the entire object mask region. For the RED-NET scene, the object mask is positioned on the left side of the images; we therefore select the left-most 512 pixels for cropping.

For the FERN scene, input images are annotated with small user-provided masks. We find that the previous approach provides too small of a mask region to the LDM's denoiser. In this case, we train RAM3D using the original dataset downsampled by a factor of 2 to a resolution of 2016×1512, and select a rectangular crop around the object mask. We compute the tightest rectangular crop which covers the mask region, and then double the crop-region height and width whilst keeping its centre intact. Finally, we increase the crop region height and width to the max of the height and width, obtaining a square crop containing the inpainting mask region. We apply this crop to the output of RAM3D and then interpolate to 512×512 before proceeding as before.

## I.5 Loss functions

During the Erase training stage, we find it necessary to backpropagate reconstruction loss gradients through pixels close to the inpainting mask (See Ablation Figure 7), to successfully reconstruct the background scene. We therefore additionally render pixels inside the Halo region (Section I.3, Figure 3), and compute reconstruction loss $\mathcal{L}_{\text{recon}}$ and perceptual loss $\mathcal{L}_{\text{vgg}}$ on these pixels, together with the corresponding region on the input images. Note that the masked image content does not fall inside the Halo region in the input images - therefore $\mathcal{L}_{\text{recon}}$ and $\mathcal{L}_{\text{vgg}}$ only provide supervision on the scene backgrounds. For the reconstruction loss, we use mean-squared error computed between the input image and RAM3D's RGB output. For perceptual loss, we use mean-squared error between the features computed at layer 8 of a pre-trained and frozen VGG-16 network [60]. In both cases, the loss is calculated on the exterior of the inpainting mask and backpropagated through the Halo region. During the Replace training phase, following [2], we apply $\mathcal{L}_{\text{BGT}_+}$ loss between our rendered output $\mathbf{x}$, and the LDM denoised output $\hat{\mathbf{x}}$, obtaining gradients to update our NeRF-scene weights towards the LDM image prior (see HiFA Loss in Figure 2, eqn 11 [2]). No other loss functions are applied during this phase, thus loss gradients are only backpropagated to the pixels on the interior of the inpainting masks. For memory and speed efficiency, RAM3D only renders pixels which lie inside the inpainting mask at this stage (Figure 3), and otherwise samples RGB values directly from the corresponding input image.

Finally, following [42], we apply depth regularisation using the negative Pearson correlation coefficient between our NeRF-rendered depth map, and a monocular depth estimate computed on the LDM-denoised RGB output. The depth estimate is obtained using an off-the-shelf model [43]. This loss is backpropagated through all rendered pixels; i.e the union of the inpainting mask and Halo region shown in Figure 3. We do not apply this regularisation during the Replace stage. In summary, the total loss function for the Replace stage is:

$$\mathcal{L}_{\text{total}} = \mathcal{L}_{\text{BGT}_+} + \lambda_{\text{depth}}\mathcal{L}_{\text{depth}} + \lambda_{\text{recon}}\mathcal{L}_{\text{recon}} + \lambda_{\text{vgg}}\mathcal{L}_{\text{vgg}} \tag{6}$$

with loss weights as follows: $\lambda_{\text{recon}} = 3$, $\lambda_{\text{vgg}} = 0.03$, $\lambda_{\text{depth}} = 3$.

We use the Adam optimiser [61] with a learning rate of 1e-3, which is scaled up by 10 for the Instant-NGP hash encoding parameters.

## I.6 Training details

Each dataset is downsampled to have a shortest image side-length (height) equal to 512, so that square crops provided to the LDM inpainter include the full height of the input image. The FERN scene is an exception, in which we sample a smaller 512 image crop within dataset images with a downsample factor of 2.

Following [13, 2], we find that classifier-free guidance (CFG) is critical to obtaining effective gradients for distillation sampling from the LDM denoiser. We use a CFG scale of 30 during the Replace stage, and 7.5 during the Erase stage. We also adopt the HiFA noise-level schedule, with $t\_min$ = 0.2, $t\_max$ = 0.98, and use stochasticity hyperparameter $\eta = 0$. In the definition of $\mathcal{L}_{\text{BGT}_+}$ loss (see eqn 11 in [2]), we follow HiFA and choose a $\lambda_{\text{rgb}}$ value of 0.1. We render the RAM3D radiance function using a coarse-to-fine sampling strategy, with 128 coarse and 128 fine raysamples. During the Replace training stage, we swap the composited background image with a randomly chosen plain

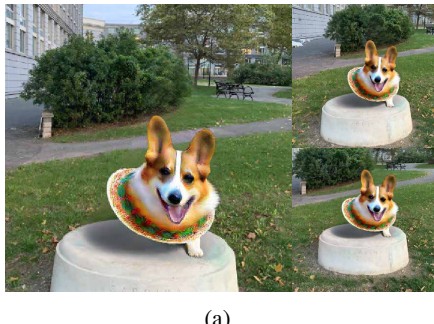
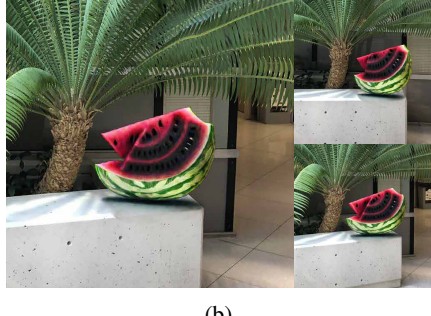

(a)                                     (b)

Figure 18: We show 2 failure cases for RAM3D. a) a challenging multi-object prompt results in an incoherent combination of both objects. b) we observe a multi-face problem which results in unrealistic geometry for the watermelon slice.

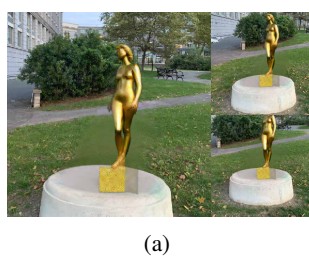
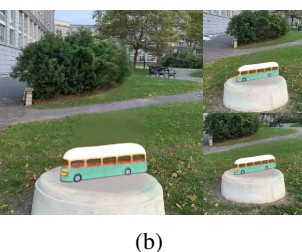

(a)                                     (b)

Figure 19: We show 2 additional failure cases for RAM3D. a): Editing object properties changes object identity. b): replacing the statue with much larger objects leads to degraded synthesis quality.

RGB image at every 3rd training step. As shown in the Ablation Figure 7, this step is critical to achieving a clean separation of foreground and background.

We train RAM3D for 20,000 training steps, during both Erase and Replace training stages. Training takes approximately 12 hours on a single 32GB V100 GPU. The output of our Replace stage RAM3D training is a set of multiview images which match the input scene images on the visible region, and contain inpainted content on the interior of the masked region which is consistent across views. To obtain novel views, we train NeRFs with standard novel view synthesis methods using RAM3D edited images and the original scene cameras poses as training datasets. We use nerf-pytorch [59] for LLFF scenes (STATUE, FERN, RED-NET SCENES), the nerf-studio [62] Nerfacto model for the FACE scene and Gaussian Splatting [36] for the GARDEN scene.

## J Limitations

As noted in Appendix 4.1, RAM3D is capable of adding multiple objects to scenes by using multiple prompts. However, we note in Figure 18 a) that adding multiple objects using a single edit prompt can sometimes result in an implausible composition which lacks a coherent spatial relationship. Furthermore, as ReplaceAnything3D is based on text-to-image model distillation techniques, our method suffers from similar artifacts to these methods, such as the Janus multi-face problem, as shown in Figure 18 b).

Our Erase-and-Replace approach might remove important structural information from original objects, so it is not suitable for editing objects' properties such as appearance or geometry (for example, turning the statue gold, see Figure 19 a).). Our method generates a Bubble NeRF from scratch inside the edit region, meaning that the structural information regarding the original statue is lost. Consequently, our model generates an entirely new statue, with gold appearance.

We note that our method works best when Erasing and Replacing objects of similar size. A significant size mismatch between Erased and Replaced objects may lead to degraded synthesis quality. This can be observed in Figure 19 b), (replacing the statue with a bus).

