# OpenReview forum: "ReplaceAnything3D: Text-Guided Object Replacement in 3D Scenes with Compositional Scene Representations"
_NeurIPS.cc/2024/Conference — NeurIPS 2024 poster_

### Official Review · Reviewer_ZFSz · 2024-06-30

**Soundness:** 4
**Presentation:** 3
**Contribution:** 1
**Rating:** 3
**Confidence:** 4

**Summary:**

This paper introduces a method that can add a novel generated or customized object into a 3D reconstructed scene. To enable this, it proposes a Erase-and-Replace strategy. The first step is to remove an object queried by a text prompt: it segment out the target object using existing language based model and fine-tune existing NeRF model based on the inpainted images. The second step is to replace the object where it fine-tune existing NeRF model using the generated or customized images. The model compared with existing instruction based gaussian splatting and NeRF methods.

**Strengths:**

- The paper is very straightforward and therefore it is easy to follow.
- The quality is convincing and promising (without considering previous works, though).

**Weaknesses:**

]The main weakness of this paper is coming from the lack of novelty.]
- The scope of the previous works highly within the previous works, and it actually does not push and advance well the boundary in the field of the research area of the 3D editing field. More specifically, it does not address any challenge which has not been addressed yet. Since the challenge is blurry, the contributions are highly suppressed.
- The proposed method is quite obvious and too straightforward. The impression from this paper, this paper is just to combine existing 3D inpainting and replacement methods. Most components inherit from previous works (or more advanced LLM components) where this paper optimize the engineering parts of the framework (e.g., effectively combine the previous modules, and better and marginally handling the text prompt, data, and mask inputs). Therefore, it also leads to the impression of the product engineering documents not research paper.
- Finally, the problem itself is not that novel and the proposed method does not even open extra-functionality in the field of 3D editing, which further suppresses the novelty of this paper.

[Potential unfairness]
- The way how existing methods use the prompt is different from the way how the authors used. Previous works have been using the prompts in a way of instruction style, e.g., turn the face to the hulk.
- To be more fair way, it would be nice if the authors can show the results from the same data with the same prompt as shown in the previous works. Since quantifying the performance of the 3D generative works is indeed challenging, this could be one of the ideal and most fair way for apple-to-apple comparison.

[Marginal quality improvement]
- Overall, the quality improvement from the previous works is not that promising. Based on the visual results, it is quite difficult to say which one is showing the better results. Based on the prediction results from the “original papers” (not this paper), it is hard to tell which ones are better. It also leads to the impression that the results shown in the paper is highly cherry-picked.

**Questions:**

Based on the current Erase-and-Replace pipeline, it is not fully clear how the authors perform only style changes, e.g., the results in Figure 5-(bottom row). During the inpainting time, if the model fully removes the objects that belong to the dedicated categories in the text prompt, there should be some artifacts such as shape distortion. However, it seems the shape of the man’s torso is perfectly preserved. In such case, how could the shape be perfectly preserved while only changing the style?

**Limitations:**

This paper discussed about the limitations.

---

> ### Author Rebuttal · Authors · 2024-08-07
>
> We are grateful to reviewer ZFSz for their review! However, we are concerned that the reviewer has misunderstood our proposed method, as the summary contains multiple factual inaccuracies. In particular, in their summary, the reviewer states that we:
> “Fine-tune the existing NeRF model based on inpainted images” during the Erase Stage, in order to remove the target object, and then “Fine-tune the existing NeRF model using generated or customized images” during the Replace stage. Both of these are misunderstandings of our method. We do not fine tune an existing NeRF scene in our method; we introduce a novel Bubble-NeRF representation (Please see Fig 3, and lines 159 to 163 in main paper) which is trained from scratch, and which models a localised NeRF region corresponding to the masked pixels.
>
> In Section 3.3 main paper, we explain how we first learn a background Bubble-NeRF during the Erase Stage, and in Section 3.4, we explain how we learn a foreground Bubble-NeRF which is composited with the background. In neither stage do we optimise these representations “using generated or customised images” as the reviewer claimed: directly training using inpainted images produces view-inconsistent results as we show in Fig 12, appendix, main paper. That is why we instead use a distillation-based training objective. However, given that the reviewer made no mention of our compositional structure, or localised Bubble-NeRF rendering in their summary (recognised as novelties by the other reviewers), we would be very grateful if they could carefully review our method (Section 3 main paper) before reconsidering their contribution rating.
>
> We now address the reviewer’s specific concerns and questions:
> “The proposed method is quite obvious and too straightforward… most components inherit from previous works…” As mentioned above, we are concerned that these statements are based on an erroneous understanding of our method. The novel components of our method; localised Bubble-NeRF rendering, Halo region supervision, compositional rendering approach were not mentioned by the reviewer in their summary. We show that an Erase-and-Replace methodology can simplify the task of 3D localised scene editing; an interesting insight that we believe the community can benefit from.
>
> “The problem itself is not that novel and the method does no open extra-functionality in the field of 3D editing” We showed how our model’s compositional structure allows adding multiple objects into the scene (Figure 6 main paper) as well as adding personalised objects using DreamBooth diffusion-finetuning in section C of the appendix, which is a practical novelty, as we are not aware of existing works with these capabilities. As far as we are aware, we are also the first to show high-quality object replacement results on 360 scenes such as the Garden scene, which constitutes a quality novelty.
>
> “The way how existing methods use the prompt is different from the way how the authors used.” Please note that we compare our method with 5 localised editing papers in our main paper (Gaussian Editor, Blended NeRF, DreamEditor, Reference-Guided Controllable Inpainting of NeRF, Repaint-NeRF). All of these methods are conditioned by a spatial prompt, indicating the region which should be edited, in addition to the target text prompt. For example, GaussianEditor specifies the editing region using segmentation masks obtained from SegmentAnythingModel, exactly like ours. The instruction-style of prompts used by InstructNeRF2NeRF (and the similar works shown in the appendix figure 10) is due to their reliance on InstructPix2Pix, an image translation method which is not required in our method. In contrast, our model is conditioned by prompts which directly describe the desired new object, exactly like GaussianEditor (Object Adding functionality), Blended NeRF, DreamEditor, Reference-Guided Controllable Inpainting of NeRF, Repaint-NeRF. We would be grateful if the reviewer could elaborate on why this is a disadvantage. Please note that InstructNeRF2NeRF tackles the related but separate problem of scene-level editing. That is why we compare our method with state-of-the-art localised editing papers in the main text of our paper, and compare with Instruct-NeRF2NeRF only for completeness in the appendix.
>
> “The quality improvement from the previous works is not that promising. Based on the visual results, it is quite difficult to say which one is showing the better results.” We have conducted a User Study which can be found in tables 3 and 4, rebuttal PDF, showing that our model’s results are preferred overall, compared with the existing works. We also show quantitative CLIP-based prompt-matching and temporal consistency results in Tables 1 and 2, which show that our model performs best overall.
>
> “It is not fully clear how the authors perform only style changes, e.g the in Figure 5 (bottom row)...” This is explained in section G2 of our appendix, main paper. As mentioned on line 571, we use a Replace-only variant of our model for the Face scene, because “the torso region in which we aim to generate new content overlaps entirely with the mask region”. Therefore, fully removing the torso in a separate Erase stage is unnecessary. Note that our method is able to generate new geometric details on the clothes (collars and jacket lapels), whilst the baseline methods keep the original geometry fixed, only updating the texture (see Figures 5 and 9 main paper).

---

> > ### Comment · Reviewer_ZFSz · 2024-08-09
> > **Follow-up questions**
> >
> > Dear authors,
> >
> > While the summary of the paper is not the major source for my decision, I am highly thankful to correct some of my understanding (fine-tuning vs. training from scratch) through the rebuttals!
> > I have a follow-up question:
> >
> > 1) I would like to confirm with the authors, the learned NeRF is still object-specific, right?
> > 2) Regarding the multi-object addition to the scene, why do you think it is not possible to perform adding multiple objects into scene using existing methods (that have shown single object insertion)?
> > 3) Questions about the prompt: I would like to confirm with the authors, do you apply the same prompt used in your method for other baseline methods?
> > 4) In the review, the quality I’ve mentioned is about the “spatial quality” which is highly correlated to the high-frequency perceptual details. Why do you think Clip-matching score can measure the spatial quality of the rendering?
> >
> > Best,

---

> > > ### Author Response · Authors · 2024-08-11
> > > **Response to follow-up questions**
> > >
> > > Many thanks for your response!
> > >
> > > 1. Yes, in the Replace stage we learn an object-specific Bubble-NeRF, which corresponds to the edit prompt $y_{replace}$ (Section 3.4, main paper). In the Erase stage (Section 3.3, main paper), we learn a scene-specific Bubble-NeRF, which represents the disoccluded background region.
> > >
> > > 2. We note that we are the first to show multi-object addition results for 3D scenes. However, we recognise that Gaussian Editor’s Add functionality can also in principle be applied multiple times. Nevertheless, as noted in our main paper lines 222 to 229, this method simply generates new objects in isolation using an image-to-3D method, which are manually positioned by the user, (see appendix F). This results in noticeable artifacts at the point of contact between the new object and its surroundings; we expect that these quality issues would be exacerbated in the multiple object case.
> > > Regarding Instruct-NeRF2NeRF and other closely related **global** scene-editing methods, these methods struggle to add entirely new objects into scenes due to their reliance on inconsistent dataset updates, as stated in the limitations section of InstructNeRF2NeRF: “adding entirely new objects to the scene (such as adding a cup on a table) is challenging for various reasons”.
> > > In comparison to existing **localised** NeRF-editing methods, note that our model differs due to its use of a compositional representation (see lines 174 - 183, and eqn 5 in our main paper). As a result of this, the bunny and strawberry results shown in figure 6 (main paper) can be represented by 2 separate Bubble-NeRFs, which are optimised iteratively. The strawberry is added to the scene first, which is then composited to become the new background model. We now apply the same process to add the bunny, ensuring scene harmonisation, whilst fixing the appearance of the strawberry. In contrast, existing localised-editing methods use a single NeRF representation for the whole scene. It is unclear whether this combined representation can be updated to insert a second object nearby to the first one as we do, without impacting the synthesis quality of the first object.
> > >
> > > 3. Yes, we use the exact same prompts when comparing with baseline methods in figure 4, figure 5 (top 2 rows), and figure 11, directly describing the new object e.g “a strawberry”. However, the Instruct-Pix2Pix based methods (figure 5 3rd row, figures 9 and 10) are conditioned by instruction-style prompts. In this case we use instruction-style prompts such as “give him a checkered jacket” for the baseline method (as described in the original IN2N paper) and “checkered jacket” for our method. Please note that whilst GaussianEditor (Replace mode) uses instruction-style prompts (3rd row, figure 5), the edits are nonetheless localised using a segmentation mask obtained from Segment-Anything, exactly as in our method.
> > >
> > > 4. We report CLIP-directional similarity to evaluate our model’s prompt adherence, following Instruct-NeRF2NeRF (and also reported by GaussianEditor, VicaNeRF, Collaborative Score Distillation, EfficientNeRF2NeRF, DreamEditor, and BlendedNeRF). Nevertheless, we have additionally conducted a User Study in tables 3 and 4 (rebuttal PDF), to further validate our results. Regarding the synthesis of high frequency details, we note that the “checkered jacket” prompt is a reported failure case in figure 9 of the InstructNeRF2NeRF paper. As noted in the figure caption, InstructPix2Pix updates “fail to consolidate in 3D” in this case, due to view-inconsistent high frequency texture details. Nevertheless, unlike the Instruct-pix2pix-based methods, our model’s distillation-based approach synthesises the correct texture, as shown in figures 5 and 9 of our paper. The challenging texture prompts “Hawaiian Shirt” and “Tartan Jacket” were similarly chosen to highlight our model’s detailed texture synthesis on the bottom row of figure 5.

---

### Official Review · Reviewer_MnXs · 2024-07-06

**Soundness:** 3
**Presentation:** 3
**Contribution:** 3
**Rating:** 5
**Confidence:** 3

**Summary:**

This paper introduces a novel method that utilizes the Erase-and-Replace strategy for text-guided object replacement in 3D scenes. Given a collection of multi-view images, camera viewpoints, text prompts to describe the objects to be replaced and to be added, this method first optimizes the background scene with the original object erased, and then optimizes the foreground object to be added. Specifically, in the erase stage, it uses the HiFA distillation technique and optimizes a localized NeRF region, which covers the mask region of the original object and the surrounding nearby pixels. In the replace stage, it optimizes the foreground NeRF within the mask of the original object while alternatively substituting the background with constant RGB tensor during the distillation. The proposed approach enables localized 3D editing and obtains more realistic and detailed edited results.

**Strengths:**

The strengths of this paper include:
(1) It proposes a novel Erase-and-Replace strategy that optimizes the background and foreground NeRF separately as compositional scenes, which obtains more realistic and detailed results.
(2) It presents extensive experiments to compare with other SOTA methods and outperforms the others in terms of the qualitative and quantitative results.
(3) The ablation study validates the effective role of the key designs of the proposed approach.

**Weaknesses:**

The limitation of the proposed approach is also obvious, as described in the limitation section. Is is only suitable for the remove, add, replace editing. The added objects should be within the pre-specified mask region. And the training is time-consuming.

**Questions:**

Questions:
(1) Figure 5 shows that the GaussianEditor makes less geometric modification compared to the proposed method. Is the main reason the Gaussian or NeRF representation?
(2) Is the proposed method able to replace the original object with a larger one? What if specifying a mask much larger than the original object?
(3) I’m curious about the property modification mentioned in the limitation. What if erasing the object with the original property and replacing it with the object with new property?
(4) It’s better to clarify the meanings of \theta_{bg} and \theta_{fg} in Section 3.2. Line 150 says “optimise RAM3D parameters \theta_{bg}”. It makes me assume that \theta_{bg} is all the parameters of RAM3D at this moment.

**Limitations:**

The paper includes a discussion about the limitations.

---

> ### Author Rebuttal · Authors · 2024-08-07
>
> We are grateful to reviewer MnXs for their review! We are glad that they find our method “novel”, that it “enables localized 3D editing and obtains more realistic and detailed editing results”, that our experiments are “extensive”, and that our “ablation study validates the effective role of the key designs”.
>
> We now address the reviewer’s specific concerns and questions:
> “The proposed approach is only suitable for the remove, add, replace editing… added objects should be within the pre-specified mask region”. This is an accurate limitation; the scope of our work is localised scene editing, on the same track as the works cited in lines 82 to 103 in the main paper.
>
> “The training is time-consuming” Our training time is in line with other NeRF-based localised scene editing approaches, but we recognise that GaussianEditor provides much faster results due to its usage of the highly efficient 3D Gaussian Splatting representation. Nevertheless, there is a clear quality tradeoff in this case as we show in figure 5 of our main paper, table 1 main paper, and the user study results in tables 3,4 of the rebuttal PDF. As we also note in figure 14 of our main paper appendix, GaussianEditor requires manual adjustment of the new inserted object’s position, unlike our method which places objects automatically.
>
> “Figure 5 shows that GaussianEditor makes less geometric modification compared to the proposed method. Is the main reason the Gaussian or NeRF representation?” We thank the reviewer for this interesting question! The main reason is the same as why Instruct-NeRF2NeRF also makes less geometric modification. Both of those methods use an iterative dataset update strategy to update the existing 3D scene (Gaussian in the case of GaussianEditor, and NeRF in the case of Instruct-NeRF2NeRF). In contrast, our method learns a localised 3D representation (Bubble-NeRF) for the edit region, from scratch. For this reason, our method is not biased towards the original scene geometry, and is able to make larger modifications.
>
> “Is the proposed method able to replace the original object with a larger one?” We tried replacing the statue with a bus in the rebuttal PDF in figure 18b). We note that the model succeeds in adding a small bus onto the plinth. However, the significant size mismatch between the Erased and newly-added object seems to lead to degraded synthesis quality.
> “What if specifying a mask much larger than the original object?”  We tried specifying a much larger mask than the original statue and generating “a super gigantic mechanical warrior robot” -  results are in the rebuttal PDF, Figure 19b). We show the outline of the original statue object in red.
>
> “What if erasing the object with the original property and replacing it with the object with the new property?”  We tried replacing the statue with a gold statue, see results in rebuttal PDF, Figure 18a). Because our method generates a Bubble NeRF from scratch inside the edit region, the structural information regarding the original statue is lost, as stated in the limitations section of the main paper on line 654. Consequently, our model generates an entirely new statue, with gold appearance - we are happy to add this figure into the limitations section in the camera-ready.
> “It’s better to clarify the meaning of \theta_{bg} and \theta_{fg} in Section 3.2…” We thank the reviewer for this recommendation, and will be happy to fix this in the camera-ready version.

---

> > ### Comment · Reviewer_MnXs · 2024-08-12
> >
> > Thanks for the response. The rebuttal has addressed my concerns. I don't have any follow-up questions.

---

> > > ### Author Response · Authors · 2024-08-13
> > >
> > > Many thanks for your useful feedback, which has helped us to improve our paper further!

---

### Official Review · Reviewer_MzLe · 2024-07-11

**Soundness:** 3
**Presentation:** 3
**Contribution:** 3
**Rating:** 6
**Confidence:** 3

**Summary:**

The authors introduce a method for replacing 3D objects within 3D scenes based on text descriptions. They do this by using in-painting approaches for the set of  images that are used for down-stream novel-view synthesis.

**Strengths:**

The paper introduces an interesting and important application of distilling 2D diffusion models into 3D.

The use of a 3D implicit representation for both the erase and replace stages is well motivated for preserving 3D consistency.

Adding a slight halo seems like a nice solution for getting slight interactions (shadows, reflections) between the generated objects and the background.

They demonstrate their method working on single and multiple objects.

The baselines they compare against seem sensible.

**Weaknesses:**

One big assumption from the way the method works is that the object you're attempting to replace is LARGER than the object you're replacing it with. SAM cannot necessarily anticipate the size difference between the two objects, and so during the erase stage the best it will be able to do is mask out the small area, in which case the area that you can inpaint over is very small. For instance, replacing the statue in Figure 2 with something larger, like a bus, will not work.

The opposite problem may also be true -- when the object being replaced is TOO much larger than the size of the object that is being added, the replace stage can end up over-sizing the object being added (perhaps encouraged by your foreground/background disentanglement, mentioned in Line 188 ). This can be seen in figure 3, where the cookies look (good but) all too gigantic in the scene.

Additionally, the Halo technique assumes that longer range interactions (further apart spatially within the image) between objects reflections don't exist, though this would easily not be true for reflective surfaces.

A table/quantitative results for Ablation studies would be more convincing. I'm especially curious about the impact of the depth reconstruction loss.

As you say in line 245, 3D scene editing is highly subjective -- in which case, I think a user study might make your claims stronger.

**Questions:**

Will the trade-offs of using BubbleNerf representation harm the final stage of training a nerf according to multi-view images synthesized by BubbleNerfs? I'm assuming that's the reason why you want to use Nerfs for your final representation, and not the BubbleNerf representations you use for the erase and replace stages. Do you trade speed/expressive power for quality when you use BubbleNerfs?

How were the language prompts chosen for objects to add?

**Limitations:**

Yes.

---

> ### Author Rebuttal · Authors · 2024-08-07
>
> We are grateful to reviewer MzLe for their review! We are glad that they found our application “interesting and important”, that our Halo region method is a “nice solution”, and that the baselines we compare to are “sensible”. We now address the reviewer’s specific concerns and questions:
>
> “Replacing the statue with something larger like a bus will not work…” We are grateful for this idea, and have generated new results for the statue scene, replacing the statue scene with a bus. As can be seen in Figure 18b) (attached Rebuttal PDF), the model adds a small bus onto the plinth. However, as the reviewer hypothesised, the significant size mismatch between the Erased and newly-added object seems to lead to degraded synthesis quality. We are happy to add this to the limitations section in the camera ready.
>
> “In figure 3, the cookies look good but too gigantic in the scene…” This is true, and is a consequence of the generative prior from the inpainting model, which tends to generate objects that are scaled to fill up the masked region as much as possible, rather than a realistic scale relative to the scene. We would be happy to add a note on this to the limitations section.
>
> “The Halo technique assumes that longer range interactions don’t exist … this would not be true for reflective surfaces.” This is true, and modelling reflective surfaces would be an interesting challenge for future work. We note that none of the existing localised scene editing papers can handle reflections either. In principle the Instruct-NeRF2NeRF-based global 3D editing papers may be able to handle reflective surfaces, but as far as we know this has not yet been demonstrated in practice.
>
> “A table/quantitative results for Ablation studies would be more convincing. I'm especially curious about the impact of the depth reconstruction loss… a user study might make your claims stronger.” We thank the reviewer for these 2 great suggestions! We have now conducted a user study, and computed additional ablation results including quantitative results, shown in the rebuttal pdf Tables 5 and 6. The ablation results include a study on the impact of the depth loss, which is important for obtaining good quality results in the “Erase” stage. We also show SAM predictions (purple region) for segmentation masks in Figure 17, Rebuttal PDF, for the ablation model variants. Note that when the depth loss is removed, the Erased statue can still be detected by SAM, when applied to our model’s results. The same issue is observed when Halo loss is removed. However, SAM cannot detect the statue region when applied to our Full model’s Erase results, and instead correctly segments the hedge behind the statue.
>
> For the user study, 15 participants were asked to compare our results with baseline models on Garden, Face and Fern scenes, using a variety of edit prompts. For each scene and prompt, participants were asked to indicate the best performing model in 2 categories. First, they were asked: “Which result most closely matches the given text prompt?”. And second, they were asked: “Which result shows the highest visual quality?” As shown in Tables 3,4 (Rebuttal PDF), our results were preferred overall in both categories.
>
> “Do you trade speed/expressive power for quality when you use BubbleNeRFs?” Thank you for this interesting question! No, there is no quality tradeoff when using BubbleNeRFs. The tradeoff is that they only represent the localised mask region in the training dataset. As a consequence, our method enables us to dedicate almost the entire GPU memory capacity to querying the relevant rays corresponding to the edit region. This allows our method to work with higher final scene resolutions than InstructNeRF2NeRF (max resolution 512) or BlendedNeRF (max resolution 168); for example, we show results on the Fern scene at a final resolution of 2016x1512.
>
> We cannot perform Novel-View-Synthesis of the entire scene using the BubbleNeRF, as it doesn’t model the scene background outside of the masked region. That is why we render our trained BubbleNeRF from every training set camera pose, and composite these renderings with the training images to obtain updated training images. These are then used to optimise a new NeRF or 3DGS scene (see Section 3.5, main paper), to obtain the final output.
>
> “How were the language prompts chosen for objects to add?” We did not use a language model to generate prompts (as was done for example in InstructPix2Pix, as it requires a large training corpus of prompts), but rather simply wrote them out manually. Similarly to diffusion-based text-to-3D works, we experimented with adding real and mythical creatures (bunnies, corgis, dragons etc) as well as inanimate objects (including foods like pancakes and cookies) as can be seen in Figure 1 main paper.

---

> > ### Comment · Reviewer_MzLe · 2024-08-13
> >
> > This addresses my original concerns, but Reviewer NhGj's followup comment is concerning -- please respond to that instead. For the same reason, I've changed my certainty to 3 (but leave the rating unchanged).

---

> > > ### Author Response · Authors · 2024-08-13
> > >
> > > We are not sure which part of Reviewer NhGj's followup comment the reviewer is referring to. If it's regarding the baseline comparisons, we have now provided the training commands and details, and point to issues posted in the ViCA-NeRF github repo indicating that other users have encountered similar problems reproducing the published results. Regarding SDS/HiFA loss vs iterative dataset updates, this stems from misconceptions regarding the InstructNeRF2NeRF IDU method, which is explained in the ablation section of InstructNeRF2NeRF.
> > >
> > > Finally, we would like to reiterate that these are all **global** scene editing methods, which we have made our best effort to reproduce, and include for completeness in the appendix (including EN2N which is not published work). **The main focus of our paper is localised scene editing,** which is why we report comparisons with the state-of-the-art localised methods in our main paper in Table 1, and Figures 4 and 5. Please let us know if there are any further clarifications we can provide to help the reviewer with their assessment!

---

### Official Review · Reviewer_NhGj · 2024-07-12

**Soundness:** 2
**Presentation:** 1
**Contribution:** 2
**Rating:** 4
**Confidence:** 5

**Summary:**

This paper studies instruction-guided scene editing. They introduce ReplaceAnything3D (RAM3D, not sure why using this abbreviation), a two-stage method to first erase the object to be replaced and in-paint the background in the scene, and then generate the replace-to object and compose it to the scene after stage one. Some relatively good visualization results are shown in the paper.

**Strengths:**

- The proposed pipeline itself is novel and well-motivated.
- Some of the visualization results look good.

**Weaknesses:**

- The method highly utilized HiFA's loss and design in their pipeline, but no ablation study was provided about HiFA to indicate whether it is HiFA or the proposed method contributes most to the good results. More specifically, I would like to understand:
    - If adding HiFA's loss and design with Instruct-NeRF2NeRF or GaussianEditor, can they achieve comparable results as the proposed method?
    - If the proposed method removes HiFA's loss and uses simple loss - so it is natural that the results are worse than the full model, will it still get better results in GaussianEditor?
- Some of the editing results are not good enough.
    - For clothes editing, there is always an obvious cutting edge (sometimes looks like the collar of a sweater) on the neck part.
- The replacement task is easier than standard instruction-guided editing, which explicitly indicates the replace-from and replace-to objects instead of requiring the diffusion model to infer.
    - This makes some comparisons unfair, especially the ones in the model that struggle to infer the replace-from region, e.g., Fig.9.
    - On the other hand, this setting also disables some editing, like more implicit instructions like "turn him into a bald" (i.e., remove hair).
    - Besides, removing an object and regenerating one prevents the model from keeping sufficient identities from the replace-from objects, which is not acceptable in some cases, e.g., color editing of an object may result in highly different objects.
    - The authors need to argue why this setting is reasonable and more/comparably reasonable compared with traditional instruction-guided editing tasks.
- The paper has multiple presentation deflects, especially the math formulas.
    - For example, the subscripts should not be italicized. $\lambda_{HiFA}$ is used in the paper, which means $\lambda_{H\times i\times F\times A}$ instead of $\lambda_{\mathrm{HiFA}}$.
    - The same variable should be referenced consistently, but the paper contains $y_{\mathrm{replace}}$ at L127/129/203 and $y_{replace}$ (which means $y_{r\times e\times p\times l\times a\times c\times e}$) at L174/185.

**Questions:**

Please see "Weaknesses".

**Limitations:**

The authors have included broader impacts and limitations in the paper.

---

> ### Author Rebuttal · Authors · 2024-08-07
>
> We thank reviewer NhGj for their review! We are glad that they found that our ‘visualization results look good’, and that the pipeline is ‘novel and well-motivated’. We now address the reviewer’s specific concerns and questions:
>
> “No ablation study was provided about HiFA” - we have now added this ablation study to the attached rebuttal PDF. Note that HiFA loss is highly related to SDS loss, and differs only by an RGB loss component (see eqn 4 in HiFA paper).  In Figure 19a), (rebuttal PDF) we show Replace-stage results using simple SDS loss (with random diffusion timestep sampling) adding a corgi in the statue scene. The results show slightly less detailed texture synthesis as the RGB-space loss is removed. Nevertheless, note that our pipeline still synthesises the new object in the correct position, which is orthogonal to synthesis quality. We additionally provide quantitative ablation results in Table 6, which show that using simple SDS loss leads to only a slight drop in CLIP Text-Image Direction Similarity.
>
> “If adding HiFA’s loss and design with Instruct-NeRF2NeRF or GaussianEditor, can they achieve comparable results?” - this is based on a misunderstanding; there is no SDS loss or other distillation loss in those methods. Instruct-NeRF2NeRF achieves scene editing using iterative dataset updates, which are obtained with a prior Instruct-Pix2Pix model. So it is not clear how HiFA loss could be incorporated into InstructNeRF2NeRF. Similarly, GaussianEditor employs the same training strategy for its Edit functionality. It differs from InstructNeRF2NeRF mainly in its choice of 3D representation (3DGS instead of NeRF). Both methods use an iterative dataset update strategy to update the existing 3D scene. In contrast, our method learns a localised 3D representation (Bubble-NeRF) for the edit region, from scratch. For this reason, our method is not biased towards the original scene geometry, which enables larger modifications.
>
> Finally, regarding the GaussianEditor object Addition functionality, this makes use of an off-the-shelf image-to-3d method (Wonder3D, which also does not use distillation) to generate the new object separately (which does not guarantee harmonisation with the original scene), and leaves it to the user to manually adjust the new object’s depth (see section F appendix main paper, and Figure 14). As we note in the main paper (line 226) this results in visible artefacts at the boundary between the object and its surroundings, and significant quality issues (see Figure 5, and User Study Table 3 in the Rebuttal PDF). It is unclear how these issues could be addressed using HiFA’s loss and design.
>
> “Some of the editing results are not good enough. For clothes editing, there is always an obvious cutting edge on the neck…” We only claim that our model outperforms the existing state-of-the-art on this task. We would be grateful if reviewer NhGj could indicate which published works can perform better, in terms of synthesising intricate localised texture/geometry patterns to match the edit prompt. We compared with the state of the art GaussianEditor in figure 5, and note that it fails to update the geometry of the clothes to match the edit prompt, whilst the texture synthesis is also inferior to our model. We would like to improve the generated geometry still further in future work, but still note that our model performs better than existing models on this challenging scene.
>
> “The replacement task is easier than standard instruction-guided editing … this makes some comparisons unfair, eg Fig.9” - Please note that in the main paper, we compare with state-of-the-art localised editing methods; Reference Guided Inpainting, Blended-NeRF, GaussianEditor, as well as DreamEditor and RePaintNeRF in the appendix. Each of these uses a spatial prompt to indicate the edit region; for example, GaussianEditor uses a SegmentAnything-based mask to select the edit region, exactly like ours. In Figure 9 (appendix) we compare with Instruct-NeRF2NeRF (standard instruction-guided editing) only for completeness - nevertheless, all main paper comparisons are like-for-like. Note that GaussianEditor (Addition functionality), Reference-Guided Inpainting, Blended NeRF, Repaint-NeRF and DreamEditor all use edit-descriptive prompts as guidance exactly like ours rather than being “instruction-guided”. We note that our Erase-and-Replace strategy for scene editing obtains better results than the existing SOTA 3D localised methods. Reducing the task to Erase-and-Replace should therefore be considered part of the appeal of our method.
>
> “The authors need to argue why this setting is reasonable compared with traditional instruction-guided editing…” Localised 3D scene editing is a well-established research track with multiple related publications (please see our Related Work section, in particular lines 82 to 103). These works may be seen as a 3D extension of 2D conditional inpainting research works such as RePaint and Blended Diffusion, and have myriad potential applications in Mixed Reality and Film Production. These applications require 3D object editing functionalities whilst keeping object surroundings consistent. Note that this is not possible for general scene-editing frameworks as they a) tend to modify the entire scene, (Fig. 9 and 10, main paper, and Fig. 3 in the DreamEditor paper) and b) struggle with object removal (Fig. 9 main paper). Therefore, the localised 3D scene editing track has emerged to address these shortcomings, and should not be considered ‘easier than standard instruction-guided editing’ as it is simply addressing a different task.
>
> “The paper has multiple presentation defects, especially the math formulas…” We are grateful to the reviewer for bringing this to our attention! We will fix these defects for the camera-ready version.

---

> ### Comment · Reviewer_NhGj · 2024-08-12
>
> I thank the authors for their rebuttal. Most of my concerns are addressed, but more concerns are raised.
>
> ***
> > “Some of the editing results are not good enough. For clothes editing, there is always an obvious cutting edge on the neck…” We only claim that our model outperforms the existing state-of-the-art on this task.
>
> I double-checked Fig.10 in the paper after receiving this response, but **more concerns** are raised about the provided results: The artifacts in baseline EN2N and ViCA-NeRF's results are *counter-intuitive*, even *unlikely* to exist in a valid setting of these baselines.
>
> - In Fig.10, EN2N generates high-quality Tartan jacket results but unreasonable results in the "checkered jacket" task. However,
>     - In Fig.9, IN2N generates failed but reasonable results of the "checkered jacket" task, at least with a clear face.
>     - Given EN2N is an *improvement* of IN2N with *same* diffusion model IP2P, I wonder **why EN2N cannot produce at least the same face-clear results as IN2N**.
> - In Fig.10, ViCA-NeRF generates two images with lots of artifacts
>     - I observed that the two images have very similarly structured artifacts, e.g., in the right images of both rows in ViCA-NeRF's results, the top-left part of the head has some hair-like artifacts.
>     - Given that ViCA-NeRF is mainly a *warp-based* method (based on NeRF-predicted depth) while both editing tasks do *not* change any geometry of the *head* part, I wonder **why similar artifacts appear in different edits** - were these artifacts already existing in the *original scene* input to ViCA-NeRF?
>
> I would like the authors to provide some analysis of these counter-intuitive artifacts. Given the code of EN2N and ViCA-NeRF is publicly available, I would also like the authors to **provide the commands that they used to obtain the results in Fig.10**, e.g., `ns-train en2n ...` or `ns-train vica ...`, so that I, along with other reviewers and AC, can reproduce these results on our side.
>
> I am paying close attention to this newly-noticed concern, since it is related to whether the baselines are compared in a fair setting, e.g., at least with reasonable hyperparameters.
>
> ***
> > “If adding HiFA’s loss and design with Instruct-NeRF2NeRF or GaussianEditor, can they achieve comparable results?” - this is based on a misunderstanding; there is no SDS loss or other distillation loss in those methods.
>
> In fact, the formulas (2)(3) in HiFA's paper have actually proven the *equivalence* between SDS loss and iterative DU's supervising rendered image with edited image. In (3) in HiFA's paper, the LHS is SDS loss, and RHS is a weighted term of $\|z - \hat{z}\|$, which is exactly the MSE loss between the latents of the original image and generated image (in one denoising step). Given that iterative DU's loss is between the original image and the generated image (in multiple denoising steps), they are actually equivalent.
>
> As we are unable to discuss on new results, I would expect the authors to provide some high-level analysis of how the techniques in HiFA, e.g., formula (4)(5)(6)(7)(8)(9) improve the results. It will be better to provide some of these ablation studies in revision.

---

> > ### Author Response · Authors · 2024-08-13
> > **Response to new concerns**
> >
> > Regarding EN2N, please note that the codebase linked by the official project site is marked as “Unofficial” - but is the only available codebase as far as we know. (Note that we are unable to post links here). We use the default command provided:
> >
> > `ns-train {en2n,en2n-small} --data <face dataset>--load-dir <nerfacto checkpoint> --pipeline.prompt {"give him a checkered jacket"} --pipeline.guidance-scale 7.5 --pipeline.image-guidance-scale 1.5 nerfstudio-data --downscale-factor 2`
> >
> > We tried both variants en2n and en2n-small (en2n-small uses a half-precision IP2P model). We obtain better results using en2n-small (shown in figure 10), while en2n completely collapsed.
> >
> > Similarly for ViCA-NeRF, we follow the default command provided in the official repo:
> >
> > `ns-train vica --data <face dataset> --load-dir <nerfacto checkpoint> --pipeline.prompt {"give him a tartan jacket"}   nerfstudio-data --downscale-factor 2`
> >
> > Regarding artefacts in the VICA-NeRF results, note that multiple users have encountered very similar issues on this dataset, as seen on the closed issues tab of the github repo. For example, please see the issues named **‘Problem of results - not the same as presented in the paper’, ‘I cannot reproduce the results :)’** and **‘Incorrect result’**. The reason why users have struggled to replicate the results is still an open question, best addressed by the authors. Nevertheless, we would be delighted if the reviewers/AC can run the code and help us (and other users!) identify the problem, and happy to update this in the camera-ready version.
> >
> > Finally, **note that these global editing results are not part of our main comparison. However, we made our best effort to reproduce them (including EN2N which is not published work), and include them for completeness in the appendix.** Please be aware of the main paper and project scope (localised editing), and let us know any questions that you have on those comparisons.
> >
> >  >  “In fact, the formulas (2)(3) in HiFA's paper have actually proven the equivalence between SDS loss and iterative DU's supervising rendered image with edited image … Given that iterative DU's loss is between the original image and the generated image (in multiple denoising steps), they are actually equivalent.”
> >
> > On the contrary, **SDS loss and IDU are quite different, as discussed in the InstructNeRF2NeRF paper, Ablation Study section**. The key difference is that SDS loss (and HiFA loss) requires rendering full images to obtain encoded latents. In contrast, IN2N training randomly samples rays across all viewpoints, which precludes VAE encoding. Therefore, incorporating HiFA into the IN2N training step would require modifying IN2N to render full images. However, this was already tried in the IN2N ablation section under the heading SDS + InstructPix2Pix. As noted, this approach “results in a 3D scene with more artifacts … We largely attribute this to the fact that the standard SDS samples rays from a small collection of full images, which makes optimization more unreliable than sampling rays randomly across all viewpoints”.
> >
> >  >  “As we are unable to discuss on new results, I would expect the authors to provide some high-level analysis of how the techniques in HiFA, e.g., formula (4)(5)(6)(7)(8)(9) improve the results. It will be better to provide some of these ablation studies in revision.”
> >
> > Please note that only equations 4 and 5 in HiFA are relevant to our method. Equations 6 to 9 are concerned with z-variance regularisation which we do not use.  Equation 3 and 4 shows how HiFA loss differs from SDS loss only by an RGB reconstruction term. As noted in the ablation section 5.2 of HiFA, the RGB term “contributes to a more natural appearance and enhanced texture details”. This section also shows that the HiFA timestep annealing scheme yields superior visual quality to standard random timestep-sampling - the reasons were analysed in section 4.1 under “a timestep annealing approach”.
> >
> > We therefore adopt both HiFA loss (equation 4 in HiFA) and timestep annealing (equation 5 in HiFA). Nevertheless, we validate this choice with the new ablation studies in the rebuttal PDF. In Figure 19a) (and Table 6), we show our model’s output using standard SDS loss and random timestep sampling. As mentioned in our rebuttal above, “the results show slightly less detailed texture synthesis as the RGB-space loss is removed”. Nevertheless, we still obtain reasonable outputs which suggests that these 2 techniques adopted from HiFA are not critical to our model’s performance. We are happy to add these new ablation studies to the camera-ready paper.
> >
> > Please note that whilst we have validated and adopted HiFAs loss function and timestep annealing as one component of our model, our core contributions are our localised Bubble-NeRF rendering, Erase-and-Replace strategy, and compositional scene representation. We hope the reviewer will take a holistic view of our work and not penalise us for using HiFA.

---

> ### Comment · Reviewer_NhGj · 2024-08-13
>
> I sincerely thank the authors for the detailed follow-up discussion.
>
> ***
> > On the contrary, SDS loss and IDU are quite different, as discussed in the InstructNeRF2NeRF paper, Ablation Study section.
>
> Again, I appreciate the authors for their detailed responses and arguments on this topic. Though I still do not fully agree with some arguments on the difference between SDS and IN2N - I also know and understand that the *practical* or *empirical* performance might be (even significantly) different, but I do not think there is a large difference in *theoretical* or *semantic* (like using different coefficients for L2 regularization may lead to success or failure in training but they do not have much difference) - I will not ask for more discussion on this point, the authors are correct that these are a little out of scope.
>
> I also appreciate the authors for mentioning which parts of HiFA to adopt in their model: _"We therefore adopt both HiFA loss (equation 4 in HiFA) and timestep annealing (equation 5 in HiFA). Nevertheless, we validate this choice with the new ablation studies in the rebuttal PDF. "_ However, I would like to point out that the point of "timestep annealing" is *never* mentioned in the main paper, while this, may contribute a lot to the generation quality. I understand that this method is widely used and is already a part of the model design, and I will not ask for an ablation study on this part. I would highly suggest the authors to add this information in the revision, to improve the reproducibility.
>
> Also, for the revision, I would kindly suggest that the authors mention fewer "adopt HiFA," "combining HiFA," and "utilize HiFA" in their revision - these may mislead the audiences (like me) that "the pipeline is highly utilizing the whole HiFA." Instead, they may directly mention the name of the components, like "HiFA's loss," and "timestep annealing strategy inspired by HiFA" to focus more on the components they adopted from HiFA, instead of HiFA itself.
>
> ***
> As for the baseline issues, in fact, these days **I have also tried to reproduce the baseline results** with all default settings:
>
> ```
> # train a nerfacto model to start, with default settings
> ns-train nerfacto --data <face dataset>
> # EN2N: default arguments of EN2N provided in their Github repo
> ns-train en2n --data <face dataset> --load-dir <nerfacto checkpoint> --pipeline.prompt "give him a checkered jacket" --pipeline.guidance-scale 7.5 --pipeline.image-guidance-scale 1.5
> # ViCA-NeRF: default arguments of ViCA-NeRF
> ns-train vica --data <face dataset> --load-dir <nerfacto checkpoint> --pipeline.prompt "give him a checkered jacket"
> ```
> And I obtained the results. As the policy about whether reviewers can post external links is not clear, I will instead describe the contents of images here. If AC permits, I will also post some images through an Anonymized Github's repo.
> - EN2N: the appearance is like IN2N, but slightly more blurred. The jacket was modified to add some scaly texture (or checkered texture with relatively small grids). I did not observe the fully unreasonable results like in Fig.10.
> - ViCA-NeRF: the appearance is that the jacket was changed to a suit with a scaly texture, while the collar part of the person has some blurred, checkered-like patterns. I did not observe any artifacts similar to those in Fig.10, either.
>
> I noticed that the authors use a different command with an additional `nerfstudio-data --downscale-factor 2`. Perhaps these are some settings inherited from IN2N or some other baselines? I am not sure whether this is the reason why the results are worse and unreasonable.
>
> In this case, I will not regard the credibility level of the results in this paper as unreliable at this point. However, I will try to reproduce the results with additional `nerfstudio-data --downscale-factor 2` during reviewer-AC discussions, and then decide whether and how to adjust the rating.
>
> I would also humbly encourage and request other reviewers that, if you have time and spare GPUs, you may also try to reproduce these results to check them from your side.
>
> Finally, again, I sincerely thank the authors for the detailed follow-up feedback. I understand that my previous comment pointed out an issue not observed in the original review and was not sent in a very timely manner, and I sincerely apologize for this. I would also suggest the authors revise their Fig.10 with the results without `nerfstudio-data --downscale-factor 2` in their revision, to achieve a fair comparison with the baselines - I agree that the proposed method is still better than the baselines in this version, and this will not change any rating about the quality.

---

> > ### Comment · Reviewer_NhGj · 2024-08-13
> > **Follow-Up Of Baseline Reproducibility Concerns**
> >
> > As a follow-up, I have completed the reproduction with the additional `nerfstudio-data --downscale-factor 2` argument. Namely, I am using the following commands:
> > ```
> > # Nerfacto
> > ns-train nerfacto --data <face dataset> nerfstudio-data --downscale-factor 2
> > # EN2N
> > ns-train en2n --data <face dataset> --load-dir <nerfacto checkpoint> --pipeline.prompt "give him a checkered jacket" --pipeline.guidance-scale 7.5 --pipeline.image-guidance-scale 1.5 nerfstudio-data --downscale-factor 2
> > # ViCA-Ne
> > ns-train vica --data <face dataset> --load-dir <nerfacto checkpoint> --pipeline.prompt "give him a checkered jacket" nerfstudio-data --downscale-factor 2
> > ```
> >
> > The results are as follows in the language description:
> > - EN2N: I tried this experiment twice, but both crashed in the middle. Before crashing:
> >     - Run 1: Though the scaly textures are also added to the jacket, the whole image was highly reddish.
> >     - Run 2: The result is like the one without `nerfstudio-data --downscale-factor 2`, but more blurred, and the scaly textures are less obvious.
> >     - Though both these runs' results do not perfectly match those in Fig.10, due the the randomness and unstability of the model itself,  it is not impossible that it may produce some results like those in Fig.10.
> >     - For academic rigor, I also reproduced the results of "Tartan jacket", and the results is quite similar to Fig.10.
> > - ViCA-NeRF: Very blurred results with flocs everywhere. These artifacts match the results as shown in Fig.10.
> >
> > From the results above, I will not regard the results as unreliable any longer. I do not know why the authors added `--downscale-factor 2` instead of using default arguments, but I tend to believe that there is a valid reason for the authors to do that, given the following facts:
> > - The original resolution of the "face" scene is 994x738.
> > - Given that IP2P was trained on a low resolution of 256x256, it looks reasonable to downscale the images to 497x369, although IP2P claimed to generalize well at 512x512 resolution (appendix A.3) and can actually work at 768 resolution (Fig.6).
> > - On the other hand, the pipeline of EN2N and ViCA-NeRF may also scale the image before inputting to IP2P, this might be the key to make it work for default settings, but may also result in failures if downscale in advance.
> >
> > Considering all these points, I would regard this issue as an unintentional misuse of the baseline codes due to an unintentional misunderstanding. In this case, I would not penalize the authors for not providing the baselines at their optimal settings in the original paper. However, the authors should still update Fig.10 in their revision for a fair comparison.
> >
> > Again, I appreciate for the detailed response from the authors. Finally, Considering all the strengths and weaknesses of this paper, I will not further decrease the ratings. At this point, I tend to keep my original rating, but I am also open to increase the rating, and may do so after engaging in the reviewer-AC discussion.

---

> > > ### Comment · Area_Chair_jSzv · 2024-08-13
> > >
> > > Dear reviewer NhGj,
> > >
> > > Thank you for your careful review of the paper and your efforts to reproduce the baselines.   Based on my understanding:
> > > - The extra `--downscale-factor 2` causes the results from EN2N and ViCA-NeRF to be degraded (similar to Fig. 10)
> > > - Without the extra `--downscale-factor 2`, the results are more reasonable.  Based on the description, it seems that the quality of the checkered jackets is not as good as the proposed methods.
> > > - There may have been some rationale for the authors to use the `--downscale-factor 2`, but a more fair comparison to the baseline would be the one without.
> > >
> > > To help me and the other reviewer's understanding, please post the images through an anonymized link.

---

> ### Comment · Reviewer_NhGj · 2024-08-13
>
> Dear AC,
>
> Thank you for following up and acknowledging my efforts. Your understanding is correct - the degrading of baseline results from an unintentional misuse (in my opinion) of `--downscale-factor 2`, while the optimal command for a fair comparison is the default one without it.
>
> I provide the results through this Anonymous Github Repo: https://anonymous.4open.science/r/neurips24_15922_review_NhGj-5684. As I do not have much time to render them as videos (also, some checkpoints are not saved due to crashing), I provide the results as Wandb validation images, where the left part is the original view and the right part is the edited view. These views are randomly sampled from all the validation views, so they might be different in different runs, but they should be sufficient for us to see the quality and artifact patterns. I apologize for the inconvenience here.
>
> Here, I would like to provide some results in images from the same viewpoint. Unfortunately, OpenReview does not support image markdowns `![]()`, and I can only provide the links. All the results except for the "Tartan jacket" one are "checkered jacket".
>
> - EN2N
>    - [EN2N Default](https://anonymous.4open.science/r/neurips24_15922_review_NhGj-5684/en2n_default/3.png)
>    - [EN2N w/ Downscaling Run1](https://anonymous.4open.science/r/neurips24_15922_review_NhGj-5684/en2n_downscale2_run1/2.png)
>    - [EN2N w/ Downscaling Run2](https://anonymous.4open.science/r/neurips24_15922_review_NhGj-5684/en2n_downscale2_run2/3.png)
>    - [EN2N w/ Downscaling "Tartan Jacket"](https://anonymous.4open.science/r/neurips24_15922_review_NhGj-5684/en2n_downscale2_Tartan/2.png)
> - ViCA-NeRF
>    - [ViCA Default](https://anonymous.4open.science/r/neurips24_15922_review_NhGj-5684/vica_default/2.png)
>    - [ViCA w/ Downscaling](https://anonymous.4open.science/r/neurips24_15922_review_NhGj-5684/vica_downscale2/5.png)
>
> It can be seen that the pattern roughly matches the ones shown in Fig.10, and there is noticible difference between w/ and w/o downscaling. Even though the results w/ default (i.e. optimal) commands are still not better than the proposed method's (proposed method produces checkered patterns in common sense, but baselines can only produce scaly textures), it is still necessary for the authors to update their Fig.10 in their revision.
>
> I hope these results could help the AC, other reviewers, and authors to understand the difference between the two settings. Thanks.

---

> > ### Comment · Area_Chair_jSzv · 2024-08-13
> >
> > Dear reviewer NhGj,
> >
> > Thank you for your quick response and providing the images.  I agree that Figure 10 should be updated with results from the original settings as they are artifact free.
> >
> > R-MzLe, R-MnXs, R-ZFSz: a quick summary
> > - There was some initial concerns regarding the artifacts in Fig. 10 which is now resolved
> > - The artifacts as been identified as due to the use of `--downscale-factor 2` when running the baselines
> > - We will request the authors update Fig. 10 with the results from the original settings, which while better than the current ones in Fig. 10, and is not really "checkered" while the proposed method produced large checkered pattern.
> > - Please check out the images provided by R-NhGj to judge for yourself
> >
> > We will discuss the paper more during the AC-reviewer discussion phase, but there is no need to be concerned about the comparison against baselines in Fig. 10.

---

> > > ### Author Response · Authors · 2024-08-14
> > > **Follow-Up Of Baseline Reproducibility Concerns**
> > >
> > > Dear reviewer NhGj,
> > >
> > > We are very grateful for your thorough investigation into these baselines!
> > >
> > > > “I do not know why the authors added `--downscale-factor 2` instead of using default arguments”
> > >
> > > We would like to clarify that **using `–downscale-factor 2` is the default argument for the given resolution.** Please note that **the official codebases specifically suggest this.** To quote from the EN2N README.md:
> > >
> > > > "**Important** Please note that training the NeRF on images with resolution larger than 512 will likely cause InstructPix2Pix to throw OOM errors. Moreover, it seems InstructPix2Pix performs significantly worse on images at higher resolution. We suggest training with a resolution that is around 512 (max dimension), so add the following tag to the end of both your nerfacto and in2n training command: `nerfstudio-data --downscale-factor {2,4,6,8}` to the end of your ns-train commands."
> > >
> > > Regarding ViCA-NeRF, the README section “Other tips for hyper-parameters” links to the InstructNeRF2NeRF repo which contains the exact same quote as above. Furthermore, on the closed github issues  “Problem of results - not the same as presented in the paper #1” and  “ I cannot reproduce the results :) #3”  which discuss quality issues when training on the exact same Face dataset, the owner of the code repository commented, recommending adding `nerfstudio-data --downscale-factor 2` to the training command.
> > >
> > > As reviewer NhGj pointed out, **“The original resolution of the 'face' scene is 994x738”**. As you can see from the above extracts from the official codebases, we simply followed the official documentation to the letter, when adding `nerfstudio-data --downscale-factor 2` to the training command. In other words, **reviewer NhGj has uncovered that the official documentation for reproducing results from EN2N and ViCA-NeRF is erroneous.** We hope that we will not be penalised for precisely following the given instructions for the baselines. Therefore, we believe that this issue should not be regarded as “an unintentional misuse of the baseline codes” but rather as caused by “erroneous documentation provided with baseline codes”.
> > >
> > > We are extremely grateful to reviewer NhGj for discovering this issue. Therefore, we will be happy to update Fig.10 in our revision, as the reviewer asked. Now that the baseline methods are generating reasonable results, the advantages offered by our proposed method are made more clear. As reviewer NhGj remarked, the updated baseline results are still not able to synthesise the correct ‘checkered’ texture pattern (unlike ours), but instead only produce “scaly textures”.
> > >
> > > > “I would highly suggest the authors add (timestep annealing) information in the revision, to improve the reproducibility ….  I would kindly suggest that the authors mention fewer "adopt HiFA," "combining HiFA," and "utilize HiFA" in their revision … Instead, they may directly mention the name of the components, like "HiFA's loss," and "timestep annealing strategy inspired by HiFA" to focus more on the components they adopted from HiFA, instead of HiFA itself.”
> > >
> > > We are very grateful for these suggestions, which we are very happy to adopt in the revision.

---

> > > > ### Comment · Reviewer_NhGj · 2024-08-14
> > > >
> > > > I sincerely thank the authors for providing more context about the command settings. The authors do not need to worry about being penalized for not using the actual best commands any longer, either it is a mislead from official instructions or an unintentional misuse of the authors.

---

### Author Rebuttal · Authors · 2024-08-07

We thank all the reviewers for their feedback! We are glad that reviewers NhGj, MzLe, MnXs found that our proposed method is novel and well-motivated, and that MzLe and MnXs in particular praised the quality of our results.

A concern that was raised by both reviewers NhGj and ZFSz was that InstructNeRF2NeRF does not use a spatial prompt to indicate the editing region, and instead uses instruction-style prompts. However, it is important to note that InstructNeRF2NeRF (together with the closely related methods shown in figure 10, appendix main paper) is a **global scene editing method**, which we only compare to in the appendix for completeness. In the main paper, we compared to state-of-art methods for **localised scene editing** (Table 1 main paper, Figure 4,5 main paper, and Figure 11 appendix), which like ours, are all conditioned with a spatial prompt indicating the 3D editing region. Furthermore, GaussianEditor (Object Adding functionality), Blended NeRF, DreamEditor, Reference-Guided Controllable Inpainting of NeRF, and Repaint-NeRF all use edit prompts which describe the new object directly (exactly as we do), rather than instruction-style prompts. The localised scene editing task has recently become a well established subtask of global scene editing, and active area of research due to its potential use cases in mixed reality applications (see lines 82-103 main paper for further details).

Reviewers MnXs and NhGj pointed out some minor presentation defects, which we are happy to address for the camera-ready version.

We provide multiple new qualitative and quantitative results in the attached rebuttal PDF document. Some reviewers proposed new experiments with variations on the input prompts and masks, for which we show qualitative results in Figures 18 and 19b).

 We ran multiple new ablation studies to validate our individual components, (on both the Erase and Replace training stages) and include quantitative results for these in Tables 5 and 6. These include new experiments testing the importance of the HiFA and depth loss terms. As in Table 1 main paper, we report CLIP Text-Image Direction Similarity for all model variants. For the “Replace” ablation studies, we used the prompt “a corgi on a white plinth”. For the “Erase” results, we evaluated using the prompt “A white plinth in a park, in front of a path”.

As an additional evaluation of the “Erase” model variants, we ran SAM on each model’s results, obtaining the main segmentation mask detected inside a bounding box around the statue region. As shown in Figure 17, rebuttal PDF (purple region), the original statue segmentation mask is still detected for our No Halo and No Depth loss model variants. However, for our full model, the statue mask is not detected; SAM instead correctly detects the hedge region behind the statue. This implies that our method has successfully removed the statue, and realistically filled in the background, including the disoccluded region of the hedge.

We also performed a User Study (as suggested by MzLe), comparing our results with 3 closely related works; Gaussian Editor, BlendedNeRF and InstructNeRF2NeRF. 15 participants were asked to compare the results from these models on a variety of scenes and prompts. In each case, they were asked to choose which result best matches the input prompt, and which one shows the highest visual quality. We report the preference rates for each model in Tables 3 and 4 (rebuttal PDF), which show that our model’s results were preferred overall across both categories.

We now address in detail the specific concerns and questions raised by the reviewers, and would welcome further comments from the reviewers seeking further clarification!

---

### Decision · Program_Chairs · 2024-09-25

**Decision:**

Accept (poster)

**Comment:**

The paper introduce ReplaceAnything3D (RAM3D), a method that allows users to replace an object in 3D scene (formed from multi-view images) by using text to describe the object to replace, and the object to add.  To accomplish this, the authors proposes a three-stage method that 1) detect and segment the object to be removed (using LangSAM), and then use 3D inpainting (LDM with HiFA) to 2) fill in the background and 3) add the requested object.  The paper includes some comparisons on selected scenes and prompts that shows the proposed method is better at replacing objects than alternative baselines.

The submission received divergent ratings with two reviewers (MnXs, MzLe) leaning toward accept, and two reviewers (NhGj, ZFSz) voting to reject.

The main concerns expressed by reviewers are:
1. Fairness of comparison against prior work
   - Missing ablation of HiFA vs key contributions of this work [NhGj]
   - Lack of systematic comparisons with same scene and same prompt [ZFSz]
   - Potential cherry picking of results (based on comparison to results show in other papers originally) [ZFSz]
2. Limited novelty with a straight-forward approach for a very specific targeted text-based editing [ZFSz]
3. Some details need to be clarified and discussion added, including:
  - clarification of specific parts of HiFA that was used [NhGj]
  - more motivation justifying the specific focus on replacement and discussion when comparing against baselines about how the baselines are more general instruction-based editing while the proposed method is designed for object replacement.
  - more discussion of the limitations regarding object size [MzLe,MnXs] and replacement when only style changed desired [NhGj,MnXs,ZFSz]
  - replace Fig. 10 with results without `--downscale-factor 2` so that generation from prior work is artifact-free [NhGj]

From the AC's perspective, the main concerns are in the fairness of comparison to prior work (1).  During the rebuttal period, the authors provided additional ablations and an user study comparing the proposed methods to baselines.  The AC believes the user study provides a reasonably systematic comparison of prior work. Reviewer NhGj also did a deep dive to investigate the artifacts shown in Fig. 10 for prior work, and determined that while the figure should be replaced to more accurately reflect the performance of prior work, the proposed method does indeed produce more accurate replacements than prior work.  Thus the AC believes concern (1) to have been mostly addressed by the author response.

For the other concerns, (3) can be addressed by improvements to the exposition, and for novelty (2) most reviewers did not have an issue with the novelty of the approach. In addition, the fact that the approach is straight-forward and intuitive does not lessen the contribution.  All reviewers also recognized the approach is very targeted, but it can be practically useful.

Due to the above, the AC believes that by incorporating the additional results and discussions from the author response period, the work would be appropriate to be presented at NeurIPS.

The authors should make sure to include the following:
- User study (from rebuttal).  It would be ideal if the the user study can be made to be more systematic (for instance, BlendedNERF and Reference-Guided Inpainting also compared against for GARDEN and FACE scenes).
- Inclusion of ablation of HiFA loss vs SDS (from rebuttal)
- Include quantitive results for ablations (from rebuttal)
- Replace Fig. 10 with results without `--downscale-factor 2` so that generation from prior work is artifact-free
- Add clarification of specific parts of HiFA that was used
- Add discussion (in experiments) of differences between the more general text-based editing that is supported by methods that are compared against vs the very specific editing supported by the proposed method
- Improved discussion of limitations, including limitations on object size, replacement when there is a style change while preserving the object shape, etc.
- Other minor improvements to writing and presentation
  - Improve notation and typesetting of math equations  (see comments from NhGj)
  - Try to clean up the references (make sure paper venues are included, consistent names for conferences, remove extra information [51], paper titles are properly capitalized).   Alphabetic ordering is recommended.